# Distributed natural gas venting offshore along the Cascadia margin

M. Riedel [1], M. Scherwath [2], M. Römer [3], M. Veloso[1,5], M. Heesemann[2] & G.D. Spence[4]

Widespread gas venting along the Cascadia margin is investigated from acoustic water column data and reveals a nonuniform regional distribution of over 1100 mapped acoustic flares. The highest number of flares occurs on the shelf, and the highest flare density is seen around the nutrition-rich outflow of the Juan de Fuca Strait. We determine ∼430 flow-rates at ∼340 individual flare locations along the margin with instantaneous in situ values ranging from ∼6 mL min$^{-1}$ to ∼18 L min$^{-1}$. Applying a tidal-modulation model, a depth-dependent methane density, and extrapolating these results across the margin using two normalization techniques yields a combined average in situ flow-rate of ∼88 × 10$^6$ kg y$^{-1}$. The average methane flux-rate for the Cascadia margin is thus estimated to ∼0.9 g y$^{-1}$m$^{-2}$. Combined uncertainties result in a range of these values between 4.5 and 1800% of the estimated mean values.

[1] GEOMAR Helmholtz Centre for Ocean Research Kiel, Wischhofstrasse 1–3, Kiel 24148, Germany. [2] Ocean Networks Canada, University of Victoria's Ocean-Climate Building at the Queenswood Campus, 2474 Arbutus Rd, Victoria, BC V8N 1V9, Canada. [3] MARUM—Center for Marine Environmental Sciences and Department of Geosciences at University of Bremen, Klagenfurter Str. 4, 28359 Bremen, Germany. [4] School of Earth and Ocean Sciences, University of Victoria, 3800 Finnerty Road, Bob Wright Centre A405, Victoria, BC V8P 5C2, Canada. [5] Present address: Universidad Andrés Bello, Facultad de Ingeniería, Calle Quillota No. 980, Viña del Mar 2531015, Chile. Correspondence and requests for materials should be addressed to M.R. (email: mriedel@geomar.de)

Natural gas emissions along continental margins were reported in many regions of the world's ocean, including active[1–4] and passive margins[5,6], shelf-seas[7–9], and the Arctic Ocean[10–12]. Often, these observations are constrained regionally or temporally, due to the nature of short-term expeditions. Understanding the natural flux of gases across the sediment/water interface is an important factor for questions of the global inventory of carbon[13,14], and associated linkages to ocean chemistry and biology[15]. In recent times, the impact of global warming on ocean chemistry and $CO_2$ uptake (ocean acidification) are key research topics[16] in conjunction with understanding similar processes in the Earth's history, such as the Paleocene–Eocene thermal maximum[17]. Additionally, a key question is how much of the methane discharged at the seafloor reaches the atmosphere and influences the global climate[18–21].

We assessed the natural gas flux (amount of gas per unit of time and area) along the Cascadia margin (Fig. 1), off the coasts of Oregon, Washington, and British Columbia (42.5°–50.5° Latitude). Understanding the geographical distribution of gas venting along the margin is closely linked to the overall tectonic process of the accretionary prism, basin development and hydrocarbon formation, erosional processes at canyons, and oceanographic phenomena (e.g., upwelling and currents) that influence the distribution of organic matter deposited on the seabed (subsequently utilized by microbes to produce methane). The occurrence of gas venting may also be linked to the distribution of rivers and estuaries providing nutrients.

The Cascadia margin has undergone subduction-related convergence at ~40 mm/y since the Eocene[22]. The shelf is underlain by a several kilometer-thick sedimentary sequence, extending from the Eel River Basin in the South to the Tofino Basin off Vancouver Island. These basins host conventional oil and gas resources indicated by petroleum exploration wells[23–25]. Natural oil seepage was reported at Barkley Canyon[26,27], and migration pathways originating from the Tofino Basin were suggested for the oil seeping out of the seafloor at the Barkley Canyon gas hydrate outcrops[28,29].

The incoming ~2.5 km thick sediment section on top of the oceanic crust consists of organic-rich hemipelagic sediments with layered turbidites[30,31]. At the deformation front, thrusts reach close to the top of the oceanic crust, and the incoming sediments are scraped off the oceanic crust and folded and faulted into accretionary ridges. An important process in the development of the prism, associated fluid flow, gas venting, and formation of gas hydrates is that of load-induced consolidation, including phase transformation of the clay mineralogy liberating fresh water[32–34] and resulting pore fluid expulsion.

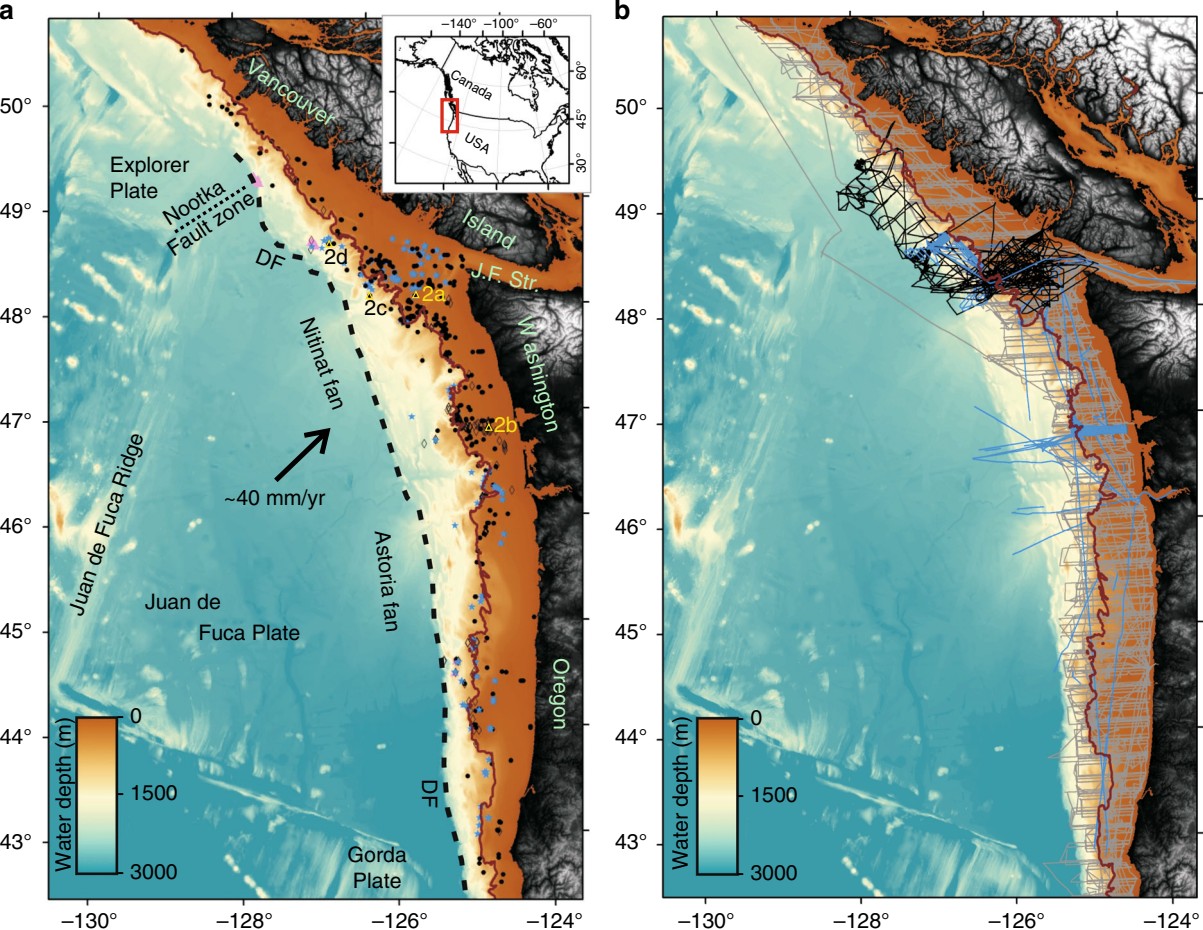

**Fig. 1** Regional maps of the Cascadia margin showing flare distribution from all available data types and line coverage. **a** Flares identified from multibeam data are shown as blue stars, black dots are from single-beam EK60 data, black open diamonds are flares reported previously, and pink triangles are from visual observations with a remotely operated vehicle. Solid brown line highlights the 500-m isobath as a proxy for the upper limit of gas hydrate stability. Open yellow triangles are examples shown in Fig. 2. Inset shows general location of study area at the west coast of North America [DF: deformation front, J.F. Str.: Juan de Fuca Strait]. **b** Complete ship track coverage used in this study: gray lines are single-beam from NOAA data base, blue are multibeam tracks, and black are all other single-beam data (Supplementary Data 1)

The widespread gas hydrate occurrences and gas venting across the prism were targeted by numerous scientific investigations[35–39], including drilling with the Ocean Drilling Program (ODP) Leg 146[40], Leg 204[41], and integrated IODP Expedition 311[42]. Assessing the amount and fluctuations in the natural flux of methane at various sites along the Cascadia margin was previously attempted[1,35–39]. However, data sets are temporarily limited and geographically sparse. Very little is known about long-term variations in the gas flux over days to weeks and months, seasonal cycles, or decades. Only one study using a sonar connected to the Ocean Networks Canada (ONC) cabled observatory node Clayoquot Slope at Bullseye Vent has been reported to date[39] showing 13 months of data. Additional evidence for changes in gas fluxes from repeat-crossings of vent regions was reported across Hydrate Ridge[36]. These and similar experiments at other margins worldwide[4,43] have indicated a tidal influence on gas emissions. Long-term studies of pore-fluid flow fluctuations at the seafloor in water depth of ~800 and ~1200 m were made at the Cascadia margin using OSMO-samplers[44–46] or the ONC crawler Wally at Barkley Canyon[47]. Although these are not directly linked to free gas discharge, they are describing the overall system and provide indications of variability over longer time scales than short-term campaign-based data. The temporal resolutions of the OSMO-samplers are comparably low with one data point every 4–6 days and correlations (other than tidal) of fluid fluxes with earthquakes or seasonal changes such as upwelling or storm patterns were investigated, but no clear correlations were recognized[44,45].

In order to better comprehend the natural gas flux and geographical distribution of gas emissions, we present a compilation of hydroacoustically determined flare sites utilizing publically available data from numerous cruises spanning 15 years (2001–2016, Supplementary Data 1). We combine different types of acoustic data such as ship-mounted 18 and 38 kHz EK60 single-beam echo-sounder, ship-mounted multibeam sonar, remotely operated vehicle (ROV)-based visual observations linked with sonar detection, autonomous underwater vehicle (AUV)-based multibeam, and long-term monitoring data from the ONC data base. Acoustic data from a total of 38 cruises were compiled and newly evaluated for evidence of gas emissions. Locations of previously reported gas vents from the scientific literature are also included[1,36,38,48,49]. We determine ~340 volumetric flow-rates at individual flare locations using the single-beam acoustic EK60 data and apply a depth-dependent density for methane gas to convert these volumetric flow-rates to mass flow-rates. A tidal-modulation function is applied to integrate the instantaneous flow-rates, estimated at the time of observation, over a 12-h tidal cycle. Two types of normalizations of the occurrences of vents are also implemented by either acoustic footprint of the data or by water depth. The margin-wide average mass flow-rate is determined to be around $88 \pm 6 \times 10^6$ kg y$^{-1}$ (with a range from ~$4 \times 10^6$ to ~$1590 \times 10^6$ kg y$^{-1}$).

## Results

**General flare identification.** Using all available acoustic data revealed a total of 914 flare locations (Supplementary Data 2, Fig. 1). We included 182 locations from previous published work[1,36,38,48,49], and added 15 locations of prolonged gas emission activity from ROV observations characterized by carbonate platforms and chemosynthetic communities. The highest number of observed flares (70%) occurs along the shelf in less than 250 m water depth (Fig. 1a). Typical examples for gas flares are shown in Fig. 2. Along the shelf, most flare locations are clustered around

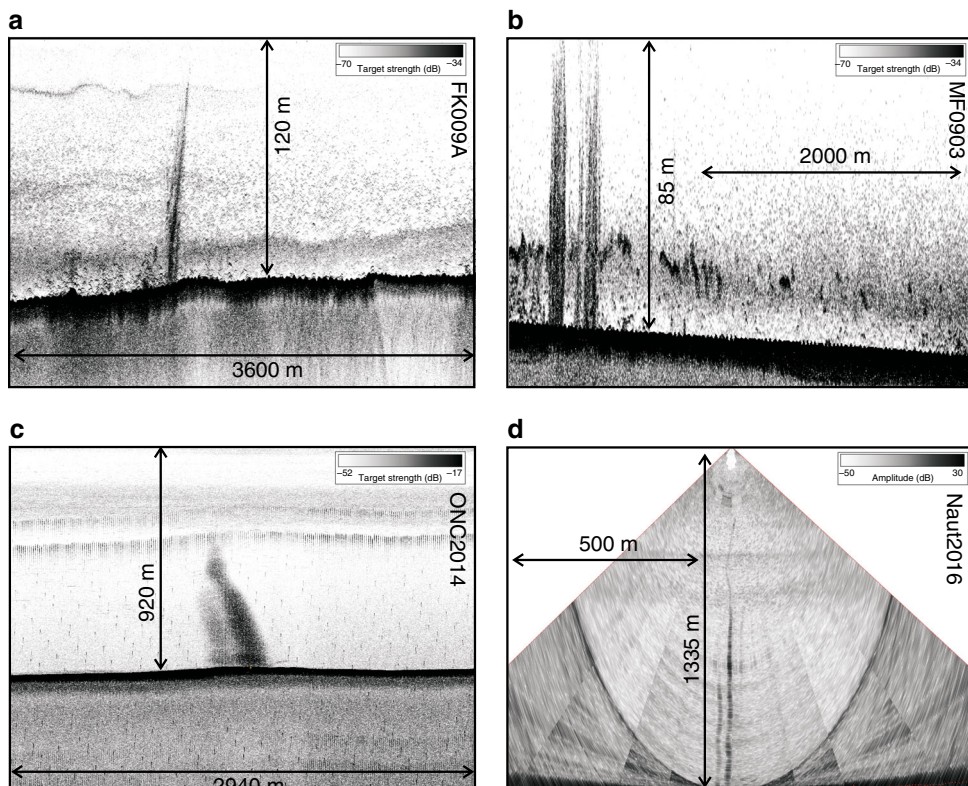

**Fig. 2** Examples of acoustic data showing gas venting. **a** Gas flare in ~120 m water depth from expedition FK009A (EK60), (**b**) example of flares in ~85 m water depth from expedition MF0903 (EK60), (**c**) example of a flare in ~920 m water depth from expedition ONC2014 (EK60), and (**d**) fan-view of EM302 data of flares in ~1335 m water depth from expedition Naut2016. Locations are shown in Fig. 1a

the entrance of the Juan de Fuca Strait and toward the rims of the Juan de Fuca and Barkley Canyon. Duplication of flare-counting between surveys was avoided by using a minimum distance between adjacent flares being larger than half of the corresponding footprint of the data used to detect the flare (see Methods on how footprint is calculated). Yet, if a flare was identified multiple times from different expeditions and different years, it provides confidence in its identification.

**The shelf environment**. Data across the shelf available from the NOAA data base were part of fishery expeditions. The cruises follow a similar regular line-pattern and extend over 10 years of acquisition (2005–2015), thus providing a reasonable view on the regional and persistent extent of gas venting. The regional coverage is, however, limited and recording depth is <750 m. Although a dense line-coverage exists along the shelf (Fig. 1b), the actual footprint of the acoustic data is small (Fig. 3d). With a typical beam-angle width of ~11° for the 18 kHz sounder, the diameter of the footprint varies between ~10 m in 50 m water depth and ~50 m in 250 m water depth. Therefore, only small portions of the shelf have actually been covered despite the high line density and thus, the number of flares is likely considerably underestimated. Extrapolation of the observed venting into uncharted areas may be misleading because flare locations are not necessarily random. They could follow geological trends or may be linked to zones of high biological productivity. Likewise,

regions without any identified flares are probably real. Since, it is overall unknown what exactly defines the occurrence or absence of flares in a specific region, predictions on the total flow-rate across the margin are carried out assuming random processes and extrapolation as outlined below.

**Deep-water setting**. Only ~30% of the identified flares are in water depths >250 m (Fig. 1a). The footprint of the acoustic data significantly increases in greater water depths (Fig. 3d), especially for multibeam data; yet, the overall line-density across the accretionary prism is smaller than on the shelf. Few surveys with acoustic data available for analyses were located across the deformation front (Fig. 1b). Combined with all available historic information gathered (Supplementary Data 1) there are possibly very few flares occurring near the trench. From all depth intervals of the prism covered, the highest number of flares (including Hydrate Ridge or Clayoquot Slope) occurs in intermediate water depths of 900–1200 m and within a distance of ~10 km landward of the deformation front. This intermediate zone has been predicated to have the highest fluid expulsion rates based on fundamental processes of porosity reduction, compaction, and associated fluid flow[34,50], which may also explain the absence of flares at the deformation front

**Normalization of depth distribution of flare locations**. Data coverage of the various expeditions is nonuniform across the

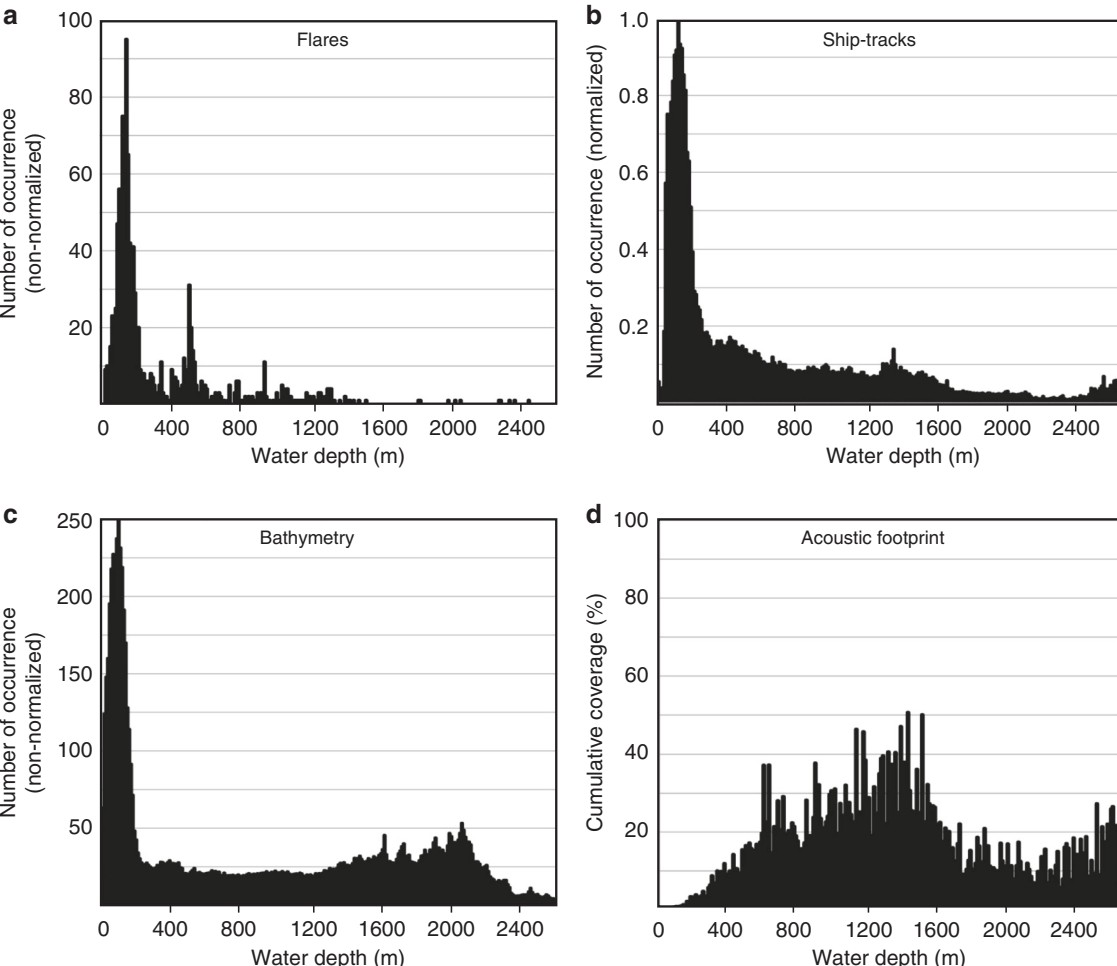

**Fig. 3** Statistical distribution of flares and data coverage. **a** Histogram of flare occurrence as function of water depth, (**b**) histogram of water depth measured along ship-tracks, (**c**) histogram of bathymetry along the Cascadia margin from 40 m water depth to deformation front, and (**d**) depth-dependent cumulative coverage (footprint) of acoustic data relative to total margin-wide depth distribution shown in (**c**)

margin. The data sets from the NOAA data base focus on shallow environments (water depth < 750 m), whereas other cruises may have followed a specific isobath (e.g., portions of expedition TN314 following the 500-m isobath), or visited a certain area repeatedly (e.g., Clayoquot Slope). Thus, normalization of the data is required. Several options exist, and we implemented two scenarios (Fig. 4): (a) following previous work[38], we used the bathymetric depth distribution of the margin from near-coast (starting at ~40 m water depth) to the deformation front (Fig. 4a), (b) using the calculated footprint area of the acoustic data at the seafloor (Fig. 4b). Normalizing the depth-distribution of flares relative to the proportions of the margin-wide bathymetry shows a maximum number of flares around the 500-m isobath (Fig. 4a). This normalization supports earlier observations of a cluster of flares at the feather edge of gas hydrate stability[37,38]. However, the data footprint increases with water depth, and when normalizing the observed depth distribution of flares by the footprint, the shelf regions become more amplified (Fig. 4b). This is a result of the shallow water environment being sparsely covered by the small beam angles of the sounders used. Cumulative areal coverage of the acoustic data increases with water depth (Fig. 3d),

and becomes 100% in some regions that have seen duplicate cruises to the same area (e.g., Clayoquot Slope).

**Acoustic quantification of gas emissions.** Mapping the regional distribution of flares is only one important element in defining margin-wide gas fluxes. Another important issue is related to flow-rates at these flares. We applied a bubble flow-rate estimation algorithm[51] (see Methods) to single-beam EK60 data. At the Clayoquot Slope site, the bulk of EK60 data across the northern Cascadia margin were acquired resulting in the most repeat observations to identify possible correlations to tides as predicted from long-term studies[39]. Here, we defined a bubble-size distribution using ROV video observations and estimated bubble rise rates for a number of bubble sizes (see Methods). The data set at Clayoquot Slope includes 114 individual flow estimates carried out for three different assumptions in the bubble rise rate[52–54], which (for identical flares) yield values within ~20% from each other. In the following discussion we refer to values based on the method by Leifer et al.[53] for clean bubbles in water depths <500 m (using this cut-off value for the regional gas hydrate stability zone, consistent with previous work[38]), and for dirty bubbles (gas hydrate coated) in deeper water depths. We define these estimates as instantaneous ($_{inst}$) in situ flow-rates, as they represent only a snapshot of the flare activities. To define a more realistic flow-rate, we implemented a tidal-modulation model (implementing the tide model driver (TMD) toolbox[55], see Methods) in which the instantaneous flow-rates are integrated ($_{intg}$) over a 12 h tidal cycle. The integration is based on a stacked tidal-forcing function extracted from the 13-month long record of bubble intensity observation[39]. In the final step, all in situ flow-rates are converted to mass of methane (kg) by applying a depth-dependent gas density[56] and simplified depth-dependent temperature profile (see Supplementary Data 3) and extrapolated to a full year (365 days).

Overall, instantaneous flow-rates estimated at all selected flare sites (Supplementary Data 3) vary between 5.6 mL min$^{-1}$ and 17.9 L min$^{-1}$ (average ~0.69 L min$^{-1}$). Individual sites that were repeatedly visited during different years and seasons showed significant variations in the flow-rate estimate (even after applying the tidal-modulation correction). These variations could stem from processes acting on longer time-scales (other than tides) as suggested by studies at Clayoquot Slope[39].

In addition to the flow-rate estimates at Clayoquot Slope, we added 310 estimates at other flare locations, 240 of those in water depths <250 m (Fig. 5). Some patterns are recognized from the regional distribution (Fig. 5) and magnitude of these estimates (Fig. 6): (a) the highest flow-rates are from flares in water depth >500 m, (b) an accumulation of relatively high flow-rates exists at the slope region up to 700 m water depth near the entrance to the Juan de Fuca Strait between 47.5° and 48.75° latitude (Fig. 5b), (c) flares in water depth <250 m show the lowest flow-rates (Fig. 6c), and ~90% of these flares show values <0.7 L min$^{-1}_{inst}$. Recognizing these differences, we defined average flow-rates (Fig. 6c) for flares in different water depth regimes to allow integrating these rates across the margin even for those locations for which no flow-rate estimate exists. The flow-rates suggest log-normal distributions (Fig. 6a, b), from which we defined average values and standard deviations. For flares in shelf water depth <250 m, the instantaneous flow-rate is ~0.1 (+0.22, −0.07) L min$^{-1}$, flares in water depth between 250 and 1000 m show flow-rates of ~0.3 (+0.74, −0.21) L min$^{-1}$, and flares in water depths >1000 m show flow-rates of ~0.7 (+2.6, −0.47) L min$^{-1}$. Applying the tidal-forcing model and integrating over 12 h yields average flow-rates of ~63 (+201, −48) L 12 h$^{-1}$ for shallow water depths, ~240

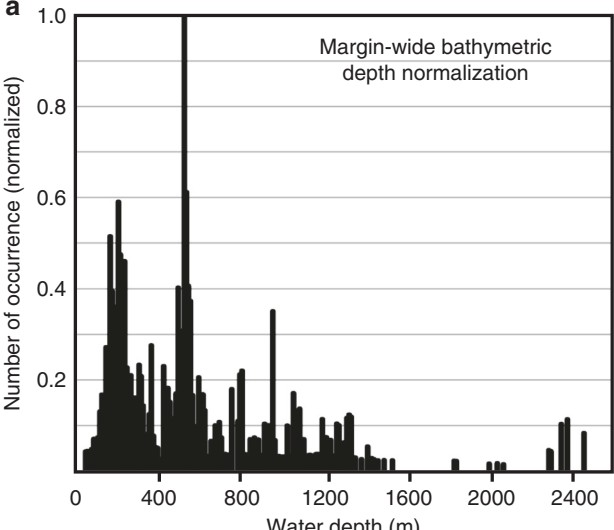

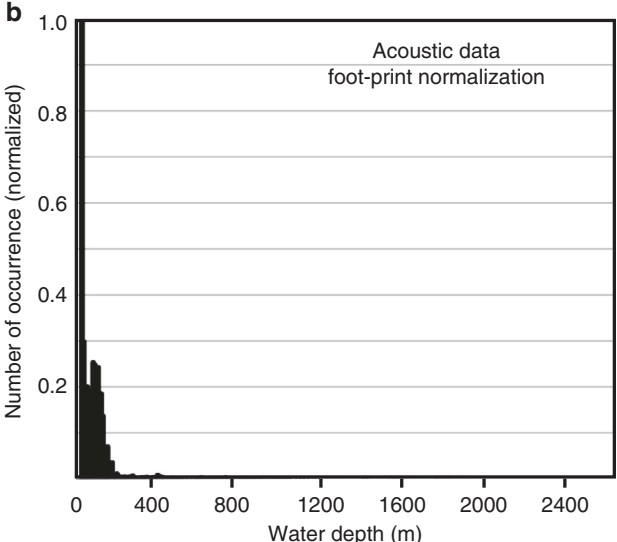

**Fig. 4** Normalized distributions of identified flares along the Cascadia margin. Shown are the normalized histograms based on **a** range in water depth of margin-wide bathymetry, and **b** footprint area of acoustic data

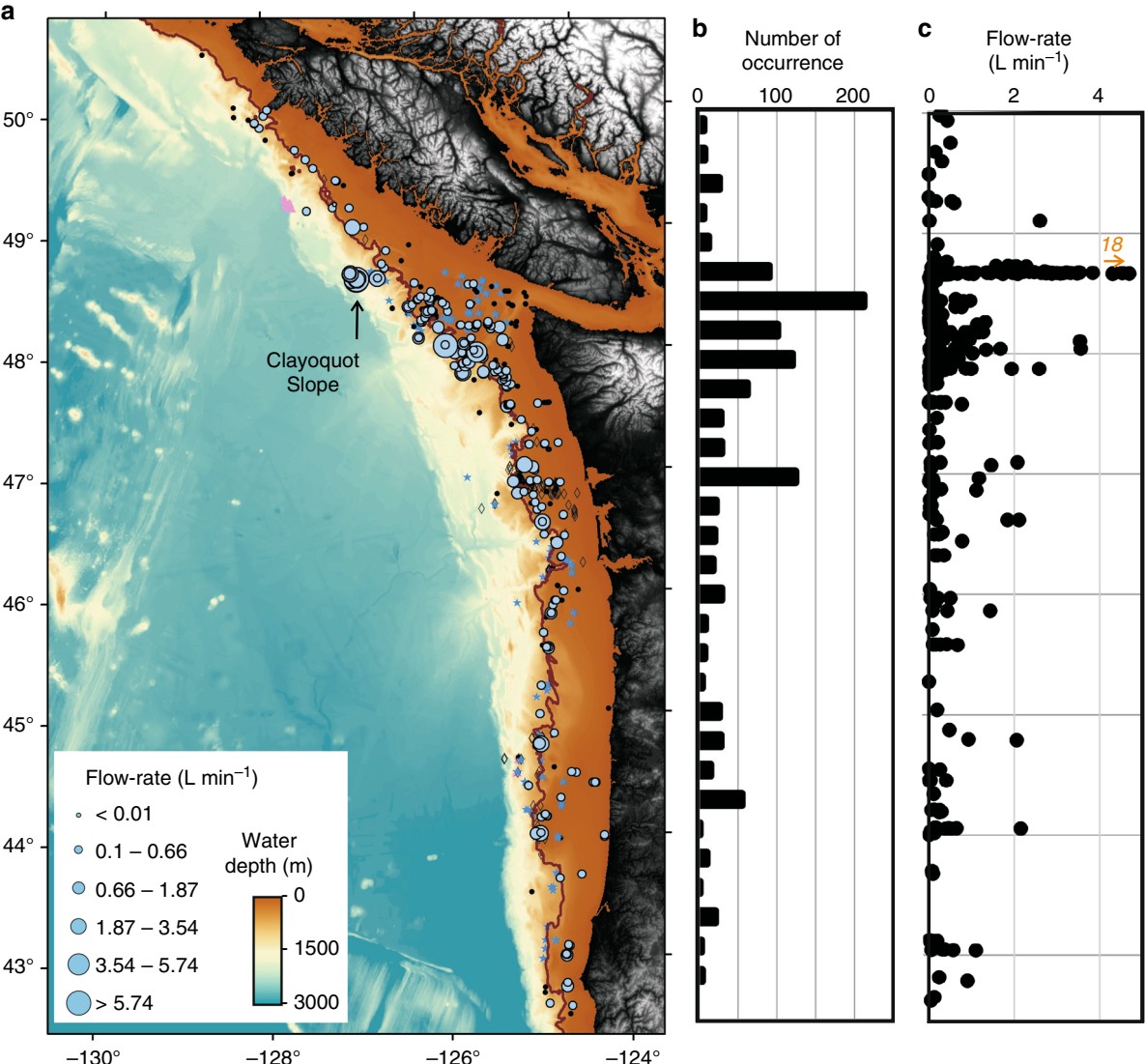

**Fig. 5** Margin-wide distributions of instantaneous flow-rates. **a** Flow-rate with symbol-size proportional to estimated values reported in Supplementary Data 3. Flares without flow-rate estimate are also shown (compare to Fig. 1). The 500-m isobath (proxy for the upper limit of gas hydrate stability) is highlighted as solid brown line. **b** Number of flares and **c** instantaneous flow-rate estimates as function of latitude

$(+850, -187)$ L 12 h$^{-1}$ for intermediate water depths, and ~624 $(+1252, -417)$ L 12 h$^{-1}$ for deep-water flares (Fig. 6c, d).

**Margin-wide gas flux**. One of the key questions we want to address is how these observations and flow-rate estimates can be combined into a margin-wide flux of methane. Several high-focus regions received intense attention over decades culminating in long-term observatories with ONC and the ocean observatories initiative (OOI). These observatories allow acquisition of longer time-series of gas emission activity and thus may ultimately allow the study of triggers or physical drivers for changes in gas emissions. Yet, these measurements will always be localized, due to the nature of the observatory settings. Assessing margin-wide flow-rates requires dense regional coverage and repeat surveys with the same acoustical tools, and validation of the acoustically defined values with measurements in situ. Although flow-rates at over 300 flares were estimated, several parameters of the calculations are yet not available everywhere. However, as a first step towards assessing the margin-wide flux of methane, we use several assumptions: on average, gas vent activity is driven by tidal

forcing; only methane venting is considered, as higher order hydrocarbons were only seen in one location and are considered exotic; no long-term modulation of the flow-rate is considered, based on the fact that our flow-rate estimates cover many years and possibly capture some aspect of this low frequency variability; gas flares in all regions are controlled by the same physical constraints and flow-rates defined are representative for all systems (i.e., identical bubble-size distribution); a systematic difference in rise-rates is applied to account for gas hydrate coating on gas bubbles for water depths >500 m; gas flares in water depths <250 m have lower flow-rates with an average of 0.1 L min$^{-1}_{inst}$ (63 L 12 h$^{-1}_{intg}$), flares in water depths between 250 and 1000 m show flow-rates of 0.3 L min$^{-1}_{inst}$ (240 L 12 h$^{-1}_{intg}$), and flares in water depths >1000 m (limited to observations < 1600 m) show flow-rates of 0.7 L min$^{-1}_{inst}$ (624 L 12 h$^{-1}_{intg}$).

The combined in situ flow-rate estimated from all the flares listed in Supplementary Data 3 is ~295.5 L min$^{-1}_{inst}$ and ~293.1 × 10$^3$ L 12 h$^{-1}_{intg}$ (average values for flow-rate estimates at duplicate locations have been used for this summation). This value contains ~50.3 L min$^{-1}_{inst}$ (47.9 × 10$^3$ L 12 h$^{-1}_{intg}$) from the

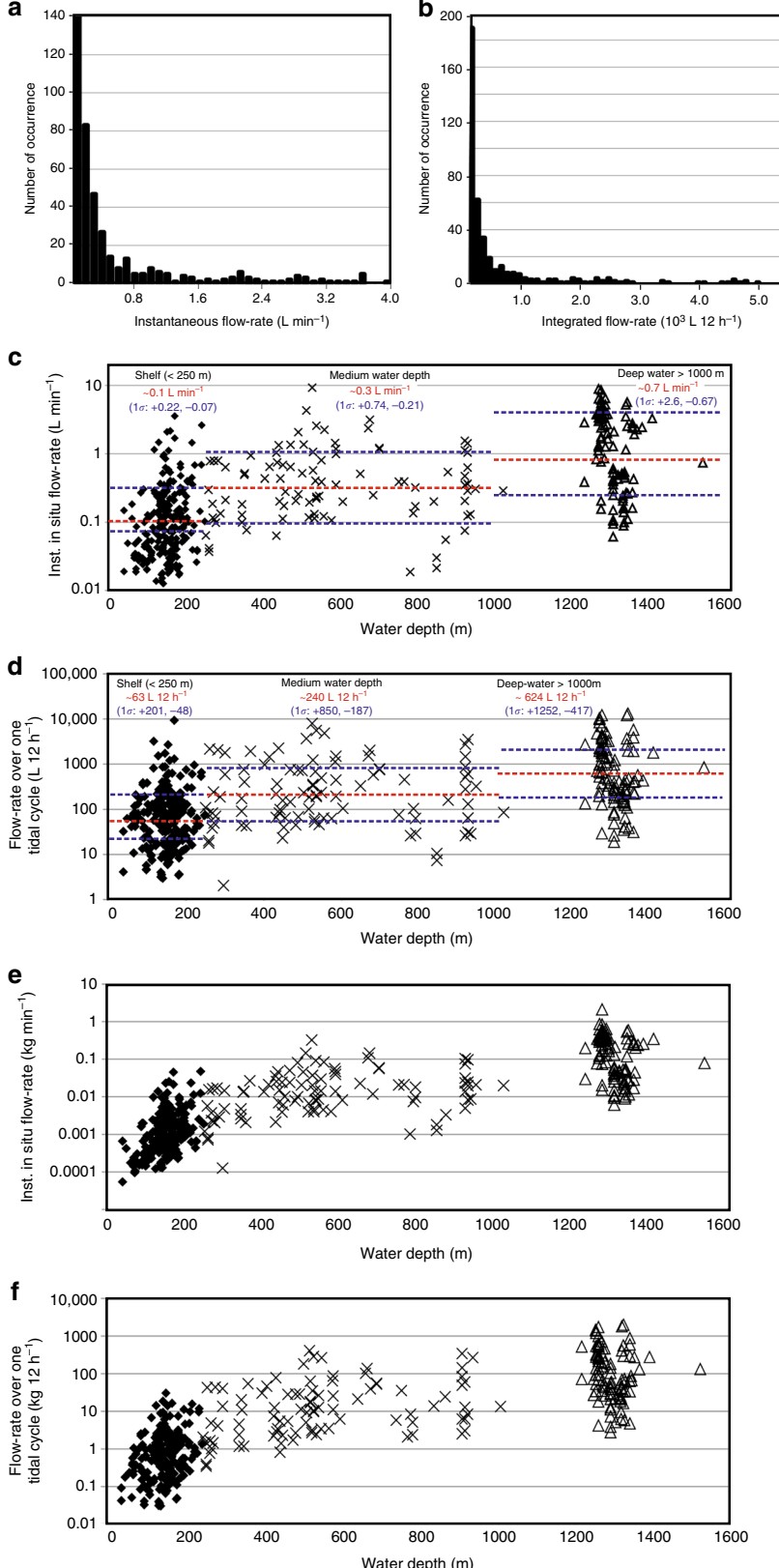

**Fig. 6** Statistics of estimated gas flow-rates and definition of average depth-dependent values used for extrapolation. Shown are histograms of **a** instantaneous and **b** integrated in situ flow-rates from all individual calculations (Supplementary Data 3). **c** Instantaneous and **d** integrated volumetric flow-rates as function of water depth. Median values are indicated by red dashed lines, and one $\sigma$ ranges are shown by blue dashed lines (see Methods), **e** Instantaneous and **f** integrated flow-rate converted to mass as function of water depth

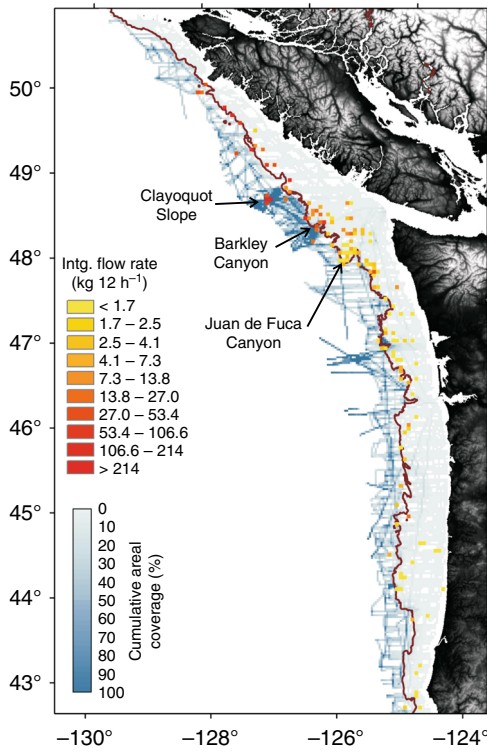

**Fig. 7** Map of integrated in situ flow-rate (in kg 12 h$^{-1}$) compared to cumulative footprint area of the acoustic data utilized. A binning process of 5 by 5 km was applied for ease of visualization. The 500-m isobath is highlighted with a solid brown line

shelf region, ~58.2 L min$^{-1}_{inst}$ (66.6 × 10$^3$ L 12 h$^{-1}_{intg}$) for water depths between 250 and 1000 m, and ~187 L min$^{-1}_{inst}$ (178.7 × 10$^3$ L 12 h$^{-1}_{intg}$) for flares in greater water depth.

Extrapolating the instantaneous and integrated flow-rates across all flare locations listed in Supplementary Data 3 over the period of 1 year yields a cumulative flow-rate of ~105 × 10$^6$ L y$^{-1}_{inst}$ (~111 × 10$^6$ L y$^{-1}_{intg}$) for these flares. Applying a corresponding depth-dependent density of methane[56], this flow-rate is equivalent to ~5.4 × 10$^6$ kg y$^{-1}_{inst}$ (~6.3 × 10$^6$ kg y$^{-1}_{intg}$). However, this value may not represent a total margin methane flow-rate as it is based on only the observed flares listed in Supplementary Data 3. Yet, it provides a realistic first-order lower bound of the flow-rate at the Cascadia margin. Thus, we combine the integrated flow-rates to visualize the regional variability in flow-rate along the margin in relation to the actual cumulative areal coverage of the acoustic data using a bin-size of 5 by 5 km (Fig. 7). This demonstrates the regional variability in flow-rates and highlights the under-representation of the shelf region with very low cumulative areal coverage (<10%).

Taking normalized flare-distributions, applying the calculated average flow-rates, and using the tidal-forcing model, a more complete measure for the margin-wide gas flow-rate may be estimated. The normalization by the depth distribution of the acoustic measurements and the margin-wide range in bathymetric depth results in similar normalized distributions. Average margin-wide in situ flow-rates after normalization with bathymetric depth are estimated to ~1848 L min$^{-1}_{inst}$ (~971 × 10$^6$ L y$^{-1}_{inst}$), or ~1.5 × 10$^6$ L 12 h$^{-1}_{intg}$ (~1100 × 10$^6$ L y$^{-1}_{intg}$), which is equivalent to ~79.7 × 10$^6$ kg y$^{-1}_{inst}$ or ~94.2 × 10$^6$ kg y$^{-1}_{intg}$. In this form of depth-based normalization, the shelf region contributes ~10% to the margin-wide flow measured in L y$^{-1}_{inst}$ (which is equivalent to a contribution of ~2% of the margin-wide flow-rate when measured

in kg y$^{-1}_{inst}$), whereas ~55% (in L y$^{-1}_{inst}$) are from the midwater range (~38% in kg y$^{-1}_{inst}$) and ~35% (in L y$^{-1}_{inst}$) are from deep-water locations (~60% in kg y$^{-1}_{inst}$). See Supplementary Data 4 for a complete list of all values and comparisons between instantaneous and integrated flow-rates.

Normalization by footprint area of the acoustic data strongly skews the distribution of flares to the shelf region (Fig. 4b). With this footprint normalization applied, the average in situ flow-rate across the entire margin is ~10,000 L min$^{-1}_{inst}$ (yielding ~5.3 × 10$^9$ L y$^{-1}_{inst}$), or ~6.5 × 10$^6$ L 12 h$^{-1}_{intg}$ (~4.1 × 10$^9$ L y$^{-1}_{intg}$), which is equivalent to ~87.9 × 10$^6$ kg y$^{-1}_{inst}$ or ~88.4 × 10$^6$ kg y$^{-1}_{intg}$. Using this footprint normalization, the shelf contributes nearly 90% of the methane flow (measured in L y$^{-1}_{inst}$), which represents ~55% of the total methane discharge measured in kg y$^{-1}_{inst}$ (see Supplementary Data 4 for more a complete list of all values). The difference between applying instantaneous or integrated values or using depth- vs. footprint-normalization results in a small spread in final average flow-rate values measured in kg y$^{-1}$ (Supplementary Data 4): (88.2 ± 6) × 10$^6$ kg y$^{-1}$. As a reference[61], the estimated contemporary global methane flow-rate across the sediment-ocean interface (in situ) is given as 16 × 10$^9$–3200 × 10$^9$ kg y$^{-1}$. Comparing global low and high estimates with our predicted average flow-rate, the Cascadia margin (with an area of ~105,000 km$^2$, representing about 0.03% of the Earth's total seafloor area) could contribute between ~0.0026 and ~0.6% of the global seafloor methane emissions.

## Discussion

Assessing the number of flares and their spatial distribution is complex and many driving forces influence the occurrence and longevity of gas emissions. Our acoustic data base spans a total of 15 years and includes data from 38 individual surveys (16 with single-beam EK60 data) with different vessels and different echo-sounder configurations. Additional historical information on gas flares was incorporated based on the available literature. Flare detection is dependent on two different sets of parameters: environmental controls and measuring techniques. Within the environmental controls, parameters such as bubble size distribution, rise rate, tidal cycle, long-term temporal variation in flow-rate, and ocean currents displacing gas in the water column are included, as well as geological controls such as association with a given sedimentary formation (i.e., natural fluid conduit), faults, ridges, canyons, or external triggers (e.g., earthquakes). A clear association with tidal forcing (start of increasing gas emission during falling tide) was recognized[39], but also phases of vent in-activity lasting weeks to months with only minor or dormant emission occurred[39]. Thus, imaging with a ship-mounted acoustic system may not recognize a particular vent if at the time of crossing either the tide is not falling and emissions are reduced, or activity is overall at a nonactive (nontidal) phase.

Mapping flares repeatedly across the margin may ultimately reveal associations of seep occurrences with some of the geological parameters. Currently, one outstanding observation is that gas emissions are clustered at two major canyons (Barkley and Juan de Fuca Canyon) and on the shelf off Grays Harbor (Washington State). This may be a result of cruise-track density, as most expeditions in our data base cross these regions to the ports of Victoria, Seattle, Port Angeles, or Grays Harbor. However, the cumulative areal coverage in these regions is not higher compared to other regions along the shelf or slope along the Cascadia margin. It is thus likely that the geographical distribution of flares is not the results of a random process or biased by cruise-track density. Several studies have shown an association between gas emissions and geomorphic features and faults or land-slide scars[57,58]. The impression that most flares occur across

the shelf around the entrance to the Juan de Fuca Strait and Grays Harbor and the upper slope regions of canyon heads may be representative of the true flare distribution. Reasons for this clustering can only be speculated upon, but may be linked to the high biological productivity of the region and thus a high abundance of organic material available for microbes to produce methane. The canyons themselves may be promoting fluid and gas flux and canyon head-scars may provide fluid escape pathways through erosion of layered sediments. The question whether flare distribution is a random process or controlled by other factors could be statistically tested, e.g., by using Horvitz–Thompson (H–T) estimators[60]. However, the data input here is a set of nonregular track lines with varying footprint and the probability of detecting a gas flare is not known a priori, which is an underlying criterion in traditional H–T estimators. Thus, for the purpose of this study, we assume that gas flares are randomly distributed and that normalization algorithms by water depth or footprint are legitimate.

The second group of parameters are highly variable and also user/operator dependent. Acoustic data to image flares span a wide range in frequency (in our case 12–38 kHz) of single-beam echo-sounders, or higher frequency with multibeam systems (in our case 12 kHz EM122, 30 kHz EM302, and 40–100 kHz EM710). Depending on water depths, bubble size distribution, bubble rise-rates, and possible turbulent flow associated with rapidly rising gas, some frequencies may be better suited for detecting gas in the water column than others. Our EK60 data base predominantly consists of 18 kHz acoustic data. Weather conditions do also affect data quality of echo-sounder data and gas flares may not be detectable if the noise level is too high. Identification of flares and separating those from other similar-looking amplitude anomalies (e.g., fish-swarms) requires training and a rigorous template of parameters that must be met to identify a flare. Single-beam echo-sounders have the tendency to smear the acoustic return from smaller vent outlets into larger-looking flares. This effect is amplified with increasing water depth. The same flares seen in single-beam data have often many smaller outlets when imaged with multibeam data or upon visual inspection with a ROV. Thus, detecting flares using single-beam data likely represents a lower limit in the number of emission sites.

The two types of normalized distribution of flares amplified the impression that most flares occur in water depths <250 m. An elevated number of flares occurs at the 500-m depth interval when using the normalization by water depth as seen previously[37,38]. However, closer inspection of this particular subset of flares and plotting their occurrence as function of geographic latitude (Fig. 5b, c) reveals that two-third of these flares are occurring between 47.5° and 48.25° latitude. Using our data set, there is no increase of vent activity along the entire margin around the water depths of the proposed gas hydrate stability feather edge. If gas hydrate dissociation has started by anthropogenic climate forcing, it should be a uniform process and not tightly constrained to a small subregion.

Using the EK60 data we estimated gas flow-rates for ~300 flares. Numerous assumptions are included, and some critical measurements are poorly known, foremost bubble size distribution and rise-rate for other vent locations than the Clayoquot Slope site. Thus, our calculated values are associated with a number of uncertainties. Totally, 30% variation in flow-rate values is defined by using different bubble rise-rate assumptions[52–54]. Differences between clean and dirty bubbles change flow-rates (for either rise-rate assumption) by slightly less than 20%. Applying our bubble-size distribution (Fig. 8) compared to other literature values from Svalbard[51] yields instantaneous in situ flow-rates that are smaller by a factor of 3. Although a depth-dependent seafloor temperature is used, a relative uncertainty remains for the actual seafloor temperatures during

the time/date of the acoustic measurement. Varying temperature within reasonable limits at flare sites changes the flow-rate by up to 5%.

Changing the tidal-forcing model and instead use extrapolation of instantaneous flow-rates (estimated in $L\,min^{-1}$) to $L\,y^{-1}$, requires multiplication with a factor of 525,600 ($60 \times 24 \times 365$). Using a depth-normalization, this calculation yields annual flow-rates that are ~90% of those using the tidal-modulation model. Interestingly, when using the normalization by the footprint area, extrapolated values of instantaneous annual methane flow-rate for all water depths along the entire margin are by ~20% higher than results using integrated flow-rates. This is likely a result of the extreme skew of values toward the shelf region. Changing the depth cut-off values from, e.g., 1000 to 1200 m water depth changes the total annual flow-rate values by <5% as only few flares were observed in this depth range.

Overall, observations from the Clayoquot Slope site[39] and other margins[4,43] suggest that gas venting is strongly tide-modulated. Thus, the simple extrapolation of the instantaneous flow-rates is most likely inappropriate; yet, as shown by previous observations[39], there are other long-term effects yet not understood that further modulate flow-rates. A measure of the overall uncertainty in the calculations can be defined by combining statistics of the range in estimated flow-rate values (which are between 25 and 370% of the respective mean values using all values from footprint- and depth-normalizations, Table 1) and uncertainty from the theory of flow-rate estimation (i.e., effects from applying different bubble-size distributions and other governing environmental constants required, which amount to 18 and 490%, respectively). Thus, the total cumulative error bounds on the average reported flow-rates are 4.5 and 1800%.

However, the geographical pattern of flow-rates may still be representative. Yet, some questions arise and require future attention: is tidal forcing of vent activity the same in all water depths? What is the bubble size distribution across the shelf? What are the long-term variations in gas venting? What controls vent location and are there geographic clusters?

Some of these questions may be addressed with long-term observatories while others require dedicated sampling efforts with ROVs, calibration of the acoustic methods with in situ flow measurements, and repeated imaging of the same regions during different tides, seasons and years. Despite the many assumptions and uncertainties in the distribution and normalization algorithms employed, we believe that our estimated total integrated annual flow-rates of ~$94.2 \times 10^6\,kg\,y^{-1}$ ($4.4 \times 10^6$–$1741 \times 10^6\,kg\,y^{-1}$) using depth-normalization, and ~$88.4 \times 10^6\,kg\,y^{-1}$ ($4 \times 10^6$–$1600 \times 10^6\,kg\,y^{-1}$) using acoustic footprint normalization are a representative first-order value for the gas flow at the Cascadia margin. Average in situ methane flux-rates for the Cascadia margin (area of ~105,000 km²) are estimated to be ~$0.9\,g\,y^{-1}\,m^{-2}$ for depth-normalization (with a range from ~0.04 to ~$16.6\,g\,y^{-1}\,m^{-2}$) and ~$0.85\,g\,y^{-1}\,m^{-2}$ using footprint normalization (with a range from ~0.04 to $15.2\,g\,y^{-1}\,m^{-2}$).

## Methods

**Single-beam acoustic EK60 water column data**. We used available data from hull-mounted EK60 echo-sounders at frequencies of 12, 18, and 38 kHz. Data from 16 different cruises with four different vessels are utilized (Supplementary Data 1). The EK60 data were displayed using the QPS Fledermaus Midwater tool and location of gas emission sites (also referred to as flares) are identified by picking the central point of the (often widespread) acoustic signal of venting at the seafloor. Individual acoustic anomalies have to meet the following criteria to be recognized as gas emissions: acoustic amplitude anomaly is connected to the seafloor; if the flare is potentially off ship track, the acoustic anomaly has to be a single, isolated, and vertically elongated stack of high acoustic energy above noise-level; acoustic amplitude diminishes with height of anomaly.

Additional indications for flares are their typical displacement within the water column by currents. Typical examples of flares seen in EK60 data are depicted in

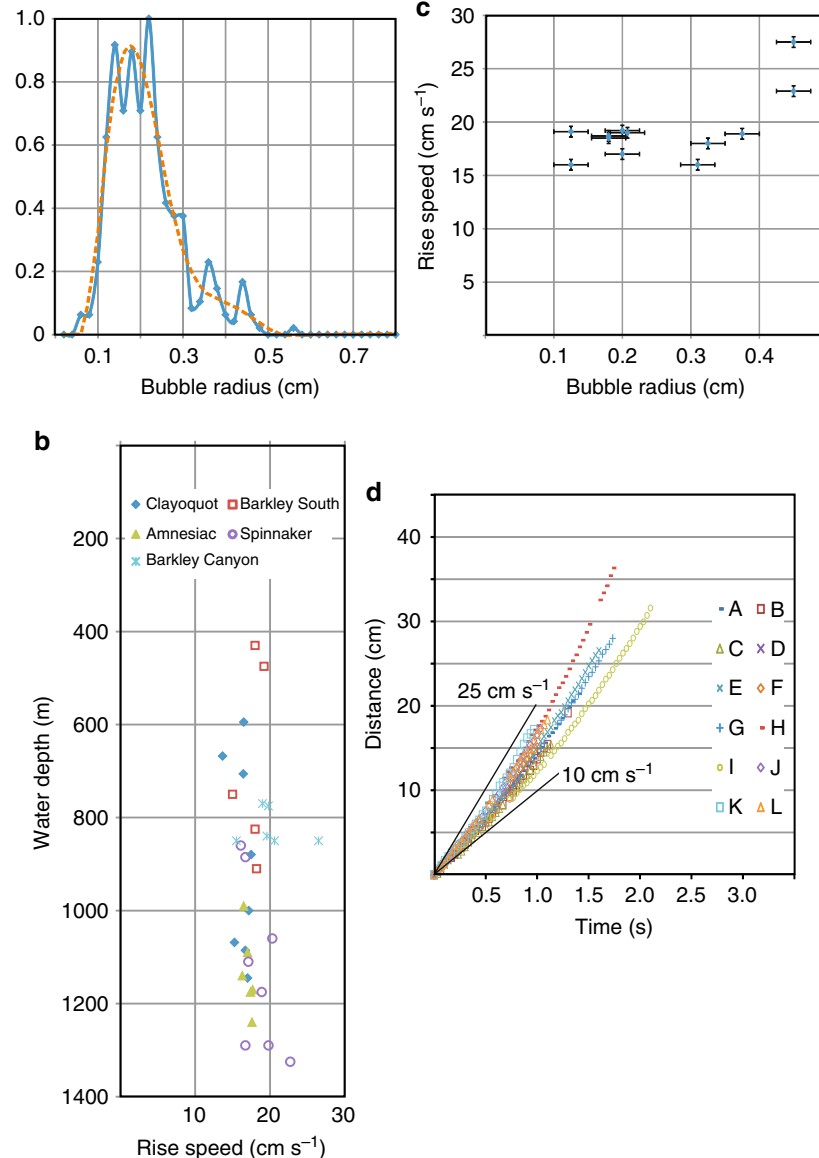

**Fig. 8** Physical parameters of gas bubbles from video observations at the Clayoquot Slope site used in the flow-rate estimation. **a** Bubble-size distribution (blue), and smoothed function (dashed orange) applied in the calculation. **b** Bubble rise rates from EK60 data over five vent locations on the Cascadia margin (for names and locations see Supplementary Data 2). **c** Selected bubble rise rates as function of bubble radius with estimated error bars from measurement uncertainty (standard deviation) of video-data. **d** Raw data of bubble tracking analysis to estimate rise rate for 12 selected bubbles (labeled A–L). Two reference slopes for rates of 10 and 25 cm s$^{-1}$ are shown as solid black lines

**Table 1 Results of extrapolation of in situ and integrated flow-rates for both types of normalization in gas flare distribution**

| Type of normalization | Instantaneous | | | | Integrated | | | |
|---|---|---|---|---|---|---|---|---|
| | Unit | Average | Minimum | Maximum | Unit | Average | Minimum | Maximum |
| Depth-distribution | L min$^{-1}$ | 1848 | 392.2 | 7143.1 | L 12 h$^{-1}$ | 1505 × 10$^3$ | 396.6 × 10$^3$ | 5.9 × 10$^6$ |
| | kg min$^{-1}$ | 155.6 | 22.4 | 637.2 | kg 12 h$^{-1}$ | 129 × 10$^3$ | 37.4 × 10$^3$ | 461.9 × 10$^3$ |
| | L y$^{-1}$ | 971 × 10$^6$ | 206.1 × 10$^6$ | 3754.4 × 10$^6$ | L y$^{-1}$ | 1098.7 × 10$^6$ | 289.5 × 10$^6$ | 4331.2 × 10$^6$ |
| | kg y$^{-1}$ | 81.8 × 10$^6$ | 11.8 × 10$^6$ | 334.9 × 10$^6$ | kg y$^{-1}$ | 96.7 × 10$^6$ | 27.3 × 10$^6$ | 337.2 × 10$^6$ |
| Footprint of acoustic data | L min$^{-1}$ | 10.08 × 10$^3$ | 2963 | 32.8 × 10$^3$ | L 12 h$^{-1}$ | 6.55 × 10$^6$ | 1.6 × 10$^6$ | 27.4 × 10$^6$ |
| | kg min$^{-1}$ | 167.1 | 41 | 599.1 | kg 12 h$^{-1}$ | 121.1 × 10$^3$ | 31.3 × 10$^3$ | 480.7 × 10$^3$ |
| | L y$^{-1}$ | 5.3 × 10$^9$ | 1557.4 × 10$^6$ | 17.3 × 10$^9$ | L y$^{-1}$ | 4.14 × 10$^9$ | 1,145 × 10$^6$ | 20 × 10$^9$ |
| | kg y$^{-1}$ | 87.9 × 10$^6$ | 21.5 × 10$^6$ | 314.9 × 10$^6$ | kg y$^{-1}$ | 88.4 × 10$^6$ | 22.8 × 10$^6$ | 350.9 × 10$^6$ |

Fig. 2a–c. Especially on the shelf region, abundant biological signals may be mistaken as flares, but we have rigorously applied these criteria to avoid false picking and cross-validated the single-beam data with coincident multibeam data. Times of venting, geographical location, and water depths are summarized in Supplementary Data 2.

**Multibeam data.** Ship-mounted multibeam systems offer a significant advantage over the EK60 system as they provide wider beam-width and thus areal coverage. Much of the margin along the northern Cascadia margin was covered in 2004 with the EM300 multibeam system, yet no water column data were recorded at the time of acquisition. All subsequent expeditions with multibeam data acquisition included the water column data (Supplementary Data 1). These data were loaded in the QPS Fledermaus Midwater tool using the stacked view as initial guide for possible venting and the fan-view for detailed detection of gas emissions (Fig. 2d). If a flare was identified, the acoustic signal was traced to the seafloor through subsequent fan-images and the geographic location was picked at the central point. The multibeam data allow also defining flare locations off the central ship track, yet identifying the outlet at the seafloor can be limited due to noise.

**Flow-rate estimation.** Acoustic detection of flares can be used to quantify the bubble flow-rate if a number of acquisition parameters and assumptions about the physics of methane gas release at the seafloor and the surrounding environments are made[51]. While acquisition parameters (such as ship's location, speed, heading, pitch, and roll, and acoustic water velocity) are usually included in the acquisition software as supplemental information in the digital raw EK60 data, the environmental variables (water temperature, salinity (35 ‰), density, and sound speed) have to be provided through additional calculations, or assumptions. Bottom water temperature in itself is strongly depth dependent and also seasonally varying, especially on the shelf[60]. Here, we adopted a simplified function (Eq. (1)) for water temperature ($T_{bot}$ in degrees Celsius) as function of depth ($D$ in meter) for the Cascadia margin, using publically available data sets from the World Ocean Circulation Experiment atlas[61]. We ignore the effect of seasonal variations and apply average values. The magnitude of variations in bottom water temperatures at the time of data acquisition is unknown.

$$T_{bot} = 0.00000076 \times D^2 - 0.00364 \times D + 6.205 \tag{1}$$

Other required parameters for the flow-rate calculations include the water properties shear viscosity ($0.0014$ Pa s), and surface tension ($0.074$ N m$^{-1}$), as well as methane gas properties of specific heat capacity ($2260$ J kg$^{-1}$ K$^{-1}$), specific heat ratio ($1.32$), thermal conductivity ($0.035$ W m$^{-1}$ K$^{-1}$), and density at sea surface ($0.656$ kg m$^{-3}$) for a given static surface pressure ($1013.25$ hPa). Density of the methane gas (in the bubbles) at the various water depths of the flare sites was calculated using the MATLAB® toolbox-algorithms developed at GEOMAR[56].

In order to relate the acoustic signals (using always a height of 10 m above seafloor) to a flow-rate, assumptions on bubble rise rate and bubble size distribution have to be made[51]. Numerous ROV video observations of flares have been made at vents off northern Cascadia, mostly close to the ONC node Clayoquot. A new bubble size distribution was defined for a flare (Fig. 8a) seen close to the bubble sonar location[39]. The bubble sizes were determined from (nonstereo and color-calibrated) ROV ROPOS video footage, where an inverted funnel with a length scale is being directly held into a bubble stream. The ROV is sitting at the seafloor during video capture at a distance of 1.5 m from the bubble stream and a camera height ~1.5 m above the funnel (effectively at an angle of 45°). The video (zoomed onto the inverted funnel) captures a height of 10–40 cm above the seafloor. Bubble sizes were measured from individual screen captures from the video and were rectified for the observation angle of the camera. Rise rates of individual bubbles were determined from a series of screen captures using the Tracker video analysis and modeling tool (version 4.9.8, available at http://www.opensourcephysics.org). One single study on the average rise rate of bubbles near Clayoquot Slope from EK60 data had been previously completed[63] with average values ranging between 16 and 21 cm s$^{-1}$. These values correspond to rise rates across a bubble size spectrum of 1–5 mm. Rise-rates over water depth from 400 to 1300 m were also newly defined for four additional vent regions off northern Cascadia using acoustic EK60 data for periods of times when the vessel was stationary (e.g., during times of water-column conductivity-temperature-depth (CTD) measurements or coring). The rise-rates defined from the acoustic data (Fig. 8b) are all close to each other (varying from 13 to 23 cm s$^{-1}$), which points to dominant bubble sizes above 2 mm diameter, which is also reflected in the video-based distribution (Fig. 8a). Rise rates were also defined from video observations for a few bubble sizes (diameters between 2 and 9 mm) and results from these sparse observations (Fig. 8c, d) generally match data and models previously published[5]. Due to the lack of a wide spectrum of direct observations in rise rate and bubble sizes (Fig. 8c), we have tested three different available literature models for clean and dirty bubble rise rates[52–54] to define the variation in flow-rate values. Estimated values for dirty bubbles are ~82% of those using clean bubbles. The method by Leifer[53] produces the lowest estimates overall (Supplementary Data 3), while the other two models yield estimates that are higher by a factor of 1.28 (Mendelson[52]) and 1.31 (Leifer and Patro[54]). Since we do not have any information

about surfactants influencing the bubble rise behavior we decided to use the method by Leifer[53] for clean bubbles for all our calculations concerning water depths shallower than 500 m, which is the approximate limit of gas hydrate stability[38]. Fluxes for sites deeper than 500 m water depth are estimated using the method by Leifer[53] for dirty bubbles to account for gas hydrate coating on the bubbles. Results using other methods[52,54] can be defined from the scaling factors mentioned above.

**AUV-based data.** AUV deployments around known cold vents in the region of Clayoquot Slope near ODP Leg 146 Site 889/890 and IODP Expedition 311 Sites U1327 and U1328 were conducted in 2009[64,65]. The AUV was equipped with a 3.5 kHz subbottom profiler and a RESON 7125 multibeam sonar. Data were converted and loaded into QPS Fledermaus Midwater tool and then treated as described above. Additional AUV data from the same expedition in 2009 were acquired across the ONC Site at Barkley Canyon and over sites of Hydrate Ridge[64]. Locations of flares identified from these expeditions are not duplicated in our compilation (Supplementary Data 2).

**ROV-based data.** Seafloor gas venting was visually observed in 2009 and 2011 with the ROV Doc Ricketts by MBARI at several seafloor vent outlets of Bullseye Vent, Bubbly Gulch, and Spinnaker Vent (all within 3 km of Clayoquot Slope) and within the coverage of the AUV mapping. The ROV was equipped with a forward-looking sonar, usually used for navigational purposes and to detect objects (or large structures) outside the visual range of the ROV (<20 m). The sonar was specifically used to identify gas within the water column in midwater depth range (~300 m above seafloor) and the anomalies found were then tracked to the seafloor for detailed bubble imaging and other analyses[63]. The AUV map indicated regions of prolonged venting from rough topography coupled with high backscatter returns[64,65]. These sites were investigated visually with an ROV and showed large carbonate outcrops and widespread chemosynthetic communities and bacterial mats[64]. However, not all of these structures are associated with gas venting in any of the acoustic data available; yet the presence of up to 2 m thick carbonate platforms exposed on the seafloor and shell debris of chemosynthetic clams point towards prolonged venting in the past. These locations have been included in our map of cold seep sites as well as earlier ROV-based observations with the ROV ROPOS in 2000 and 2001 of carbonate formations and chemosynthetic communities around Bullseye Vent and other carbonate outcrops[66]. Outside the northern Cascadia accretionary prism of the Juan de Fuca Plate, one additional region with evidence for venting has been identified by ROV observations. At the foot of the Nootka slope region, several ROV dives with ROPOS made to install seismic monitoring equipment[67,68] also showed chemosynthetic clams, tube worms, bacterial mats and thick carbonate concretions. However, no acoustic evidence for gas venting has been seen in any of the surveys covering that portion of the slope. A long history of ROV observations for studying seafloor outcrops of gas hydrate, carbonates, and chemosynthetic communities exist at the southern Cascadia margin, especially around Hydrate Ridge[48,64,69]. New long-term monitoring equipment for studying these outcrops are currently installed at the OOI sites. Locations from these previous publications have been incorporated (Fig. 1).

**Tidal model and impact on flow-rate estimate.** In order to define the time of acoustic data measurements relative to the occurrence of ocean tides, we have implemented a tidal model based on bathymetric data available for the region off the Pacific West Coast and the TMD toolbox[55]. The tides were calculated using the following constituents: M2, S2, N2, K2, K1, O1, P1, and Q1. For each of the EK60 data points used to estimate flow-rates, the time of measurement and location (latitude and longitude) was used to define a tidal time series of 15 h prior to each data point from which the closest previous low-tide was defined. The time difference from the low-tide and the data measurement is reported in Supplementary Data 3. We further used the available time series from the bubble sonar at the Clayoquot site[39] to define an average function of flow-rate variation throughout a tidal cycle. As previously described[39], the vent activity is strongly modulated by the tides with maximum intensity of gas venting occurring ~5 h after the low tide. The individual flow-rate estimates shown in Supplementary Data 3 occur randomly distributed in time across a tidal cycle (Fig. 9a). We stacked a total of 40 tidal cycles, normalized to the maximum observed bubble activity during each tidal cycle (Fig. 9b), to define a vent forcing function (f(t), with t denoting time in hours). The forcing function (Fig. 9c) was then fitted with a three-term Gaussian polynomial:

$$f(t) = a_1 \times \exp\left(-((t - b_1)/c_1)^2\right) + a_2 \\ \times \exp\left(-((t - b_2)/c_2)^2\right) + a_3 \times \exp\left(-((t - b_3)/c_3)^2\right). \tag{2}$$

The coefficients for Eq. (2) are defined in Supplementary Data 5. The integral of this function over a total of 12 h then defines the volume of gas venting occurring over a complete tidal cycle. For a normalized forcing function with maximum intensity occurring at 5 h after the previous low-tide, the integral value over a complete tidal cycle is defined to 3.42. To calculate the maximum potential flow-rate during a tidal cycle of a particular vent, the estimated flow-rate (in units of L h$^{-1}$) at a given time is divided by the corresponding value of the vent forcing function. Finally, this maximum possible flow-rate is multiplied by the integral

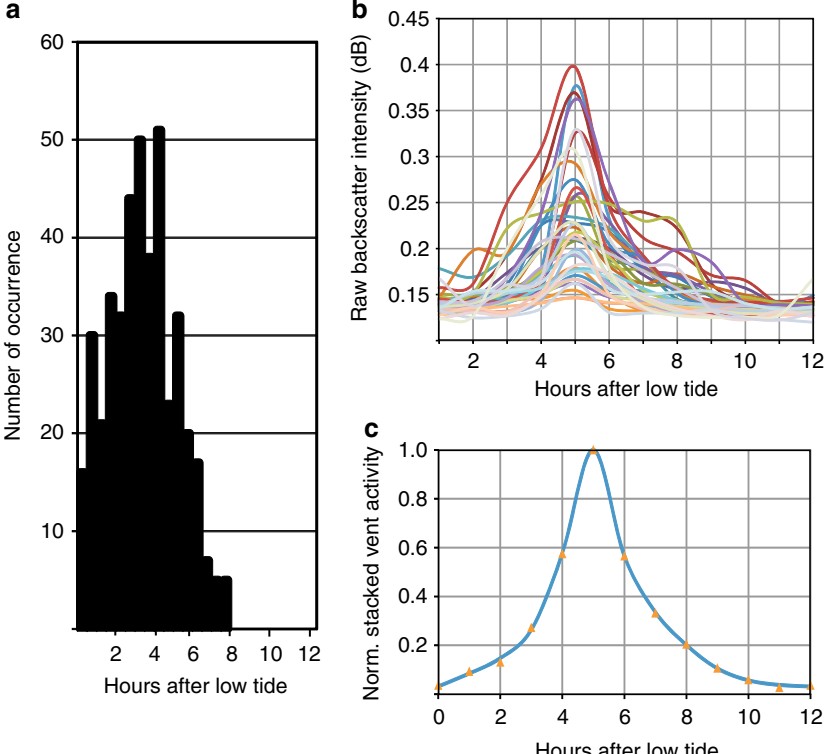

**Fig. 9** Definition of the tidal-forcing function. **a** Histogram of number of occurrence of flow-rate estimates relative to time after low-tide. **b** Example of 40 vent activities defined from bubble sonar[39] as function of time over a tidal cycle. **c** Stacked and normalized vent force function with orange symbols as stacked values and blue line from Eq. (2) of a three-term Gaussian best fit function

value of the normalized forcing function to yield the total flow-rate over a twelve hour long tidal cycle for each vent.

**Normalization of vent distribution**. Data acquired at the Cascadia margin are not normally distributed (Fig. 1b). Most of the acoustic surveys are across the shelf (e.g., from the NOAA data base). Only few of the scientific expeditions ventured to the toe of the accretionary prism. Thus, detection and counting the occurrences of vents needs to be normalized by the distribution of the acoustic data. Two options exist for normalization: using the range of observations per water depth and using actual footprint area of the acoustic data at the seafloor, which is in itself dependent on water depth. In order to achieve a measure of the data distribution in depth and to define the footprint area, all original cruise navigation was utilized. For each of the acoustic measurement points within the shelf and slope region to the deformation front (but excluding the entry-way of the Juan de Fuca Strait) we defined water depths from the same seafloor bathymetric reference grid[70]. A histogram of the depth values spaced at 10-m bins for a range in water depth to 3400 m was determined to yield values ranging between zero (no data point) and one (maximum number of data points).

The footprint area was first defined on a single ship-track basis using the bathymetric reference grid[70] and the individual EK60 beam angle information per vessel and cruise (Supplementary Data 1). Overlaps between individual consecutive acoustic data EK60 points from the same track line within a specific cruise were excluded from the footprint calculation. For multibeam cruises, an effective opening beam angle of 26.5° was used for calculating areal coverage, as not the full 60° swath illuminated seafloor can be used for detection of acoustic flares (see Fig. 2d for an example of multibeam data coverage across a flare). We further simplify the area imaged acoustically by a single multibeam swath per navigation point to a rectangle, where the horizontal length is defined by water depth and the effective opening beam angle. The ship tracks along lines with multibeam data acquisition were decimated into equal-distant points with a 100-m step-size. Thus, the rectangular area $A$ covered by a single multibeam swath per decimated navigation point is approximated using the simple expression $A = 100 \, \text{m} \times \text{water depth}$. Overlap between different track lines either from the same or other cruises in different years was included in the integration of all footprint area values. The footprint area information was gridded using Esri ArcGIS (Version 10.2) into a raster with 2.5 km × 2.5 km grid cells (Fig. 7). A histogram of average footprint area for each 10-m depth bin was defined by using the average coverage in each depth bin divided by the total area this depth bin represents along the entire margin (over the range from coast to deformation front and within our latitudinal limits, i.e., represented by the histogram of Fig. 3c).

The cut off values for shelf (<250 m), medium (250–1000 m), and deep (>1000 m) water depth settings were defined using the bathymetric data and range of acoustic data available. Along Cascadia, the shelf break (defined as location of significant change in slope angle) coincides on average with the 250-m isobath. The definition between medium and deep-water setting is defined from the spread in acoustic and flow-rate estimates (Fig. 6). A gap of observations between 1000 and 1200 m water depth exist, with flares in deeper than 1000 m water depth showing significantly higher flow-rate values. The extrapolated flow-rate values are reduced by <10% if the cut-off value is shifted from 1000 to 1200 m.

**Uncertainty estimation**. The estimation of flow-rate at any given flare as well as the regional extrapolation are inherently uncertain, and includes theoretical assumptions, data scatter, and model simplifications. We outline in the following our approach to define an overall uncertainty in the reported values of flow-rates and margin-wide fluxes, summarized in Table 1.

The method used in this study to calculate flow-rate[51] requires many input parameters as described above. Some of these parameters do change between individual flare sites, others are physical constants. The following parameters were used to define the overall range in uncertainty in flow-rate estimation: water depth, seafloor temperature, salinity, sound speed, and density, as well as assumptions made on bubble size distribution and applied bubble rise rate model. We defined measures of uncertainty in the flow-rate empirically for the flare sites by varying only one input parameter at a time, holding all others at constant values.

The bubble-size distribution may vary between sites and regions[51,53,54] and flow-rate estimation can thus vary when applying the different distributions on the same acoustic data set. We have used a different bubble size distribution from off Svalbard[51] for the entire set of acoustic flares reported in Supplementary Data 3 (while holding all other parameters constant) to evaluate the magnitude of the uncertainty from this parameter in the calculations. The flow-rates from all flares are on average higher by a factor of three, compared to results when using the bubble size distribution shown in Fig. 8a. While this is not an exhaustive comparison and not mathematically defined, it shows a general trend linked to the percentage of large bubble-sizes (>0.3 cm radius): if the bubble size distribution contains a higher abundance of larger bubbles, the flow-rate goes up (and vice versa).

The overall uncertainty of the flow-rate estimation based on the applied physics is defined as a simple superposition (multiplication) of individual factors of uncertainty: degree of uncertainty in seafloor temperature (Eq. (1)) and its effect on flow-rate was found to be maximum 5%, with lower temperatures generally reducing the flow-rate; degree of uncertainty in near seafloor salinity from deep ocean to shallow water shelf environments (1–2‰), and its effect on flow-rate was

indiscernible; degree of uncertainty in salinity and temperature on seawater density and sound speed and impact on flow-rate estimation was found to be indiscernible; application of dirty versus clean bubble models reduces flow-rate by ~18%; application of different bubble rise rate models[52–54] changes flow-rate by a factor of ~1.3; application of different bubble size distributions changes flow-rate estimate on average by a factor of 3.

Therefore, we propose a total uncertainty in the theory of flow-rate estimation described by a factor of ~4.8 (by which any flow-rate could be higher) or by a factor of ~0.18 (by which any flow-rate could be lower). In other words, a flow-rate supposedly determined to 100 mL min$^{-1}$ could be as low as 18 mL min$^{-1}$ or as high as 480 mL min$^{-1}$.

On top of this method-based uncertainty lies the scatter of our input data (as represented in Fig. 6c, d). Uncertainty from the scatter of the input data and impact on regionally extrapolated flow-rates for all flares at the Cascadia margin is dealt with by using regional average values and associated standard deviations and reporting of average possible flow-rates with ranges derived from the log-normal distributions (Table 1). The flow-rates reported in Supplementary Data 3 are first transformed into the log-domain. From those log-values (which are now normally distributed), the log-mean and log-standard deviations are derived. The log-mean is equivalent to the median in the original data-domain[71]. In order to define the possible ranges in flow-rates, the log-standard deviations are back-transformed into the data-domain and the minimum and maximum values are now unsymmetrically distributed around the median[71].

A last level of uncertainty is introduced from the treatment of the regional binning of seafloor depth, echo-sounder footprint calculation, and normalizations applied. The echo-sounder footprint calculation is based on several assumptions related to the sounder beam angle and roughness of the seafloor. At each step of the footprint calculation along any given ship track, we assume a flat seafloor. This is a reasonable assumption on the shelf, but possibly inaccurate in larger water depths on the slope with increased topography. However, with a binning size of 5 by 5 km and defining a cumulative footprint within this bin from multiple surveys (and subsequent normalizations), the error from inaccurately defined seafloor roughness is deemed negligible.

**Data availability**. The acoustic water column data (EK60 and multibeam) utilized in this study are available through data sources defined in Supplementary Data 1 for each cruise. These include the online-accessible data portals of NOAA, Ocean Networks Canada, and Natural Resources Canada (Geological Survey of Canada). Data used to constrain flow-rate estimation of flares (e.g., seafloor temperature and pressure) are based on long-term data made available through the Ocean Networks Canada portal.

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

## Acknowledgments

Our work and analyses presented in this study are built around various data sets acquired over more than 15 years, by many different scientific teams, and during many individual research expeditions to the Cascadia margin. Our thanks go to all scientists and crew involved in data collection. Additionally, we like to thank Charles Anderson and Carrie Wall, NOAA Affiliates, who provided substantial help in accessing and downloading the EK60 and multibeam data from the NOAA data-base.

## Author contributions

M.R, M.S., and G.D.S. compiled all acoustic data. M.R. performed flow-rate estimation and defined all statistical elements used in the analyses. M.V. contributed to the flow-rate calculations. M.H. contributed input to tidal prediction model, and M.R. provided long-term observation data at Clayoquot Slope. M.R. wrote the manuscript with contributions from all the coauthors.

## Additional information

**Competing interests:** The authors declare no competing interests.

