## [Peer Review File · Nature Communications]

Reviewers' comments:

Reviewer #1 (Remarks to the Author):

Overview

The manuscript "Distributed natural gas venting along the Cascadia Margin" presents a compiled database of methane vent locations on the Cascadia Margin, as well as associated flux estimates for select vents, derived from archived acoustic water column backscatter data and previous publications. The authors synthesize the results to provide a summary of vent distribution patterns (both horizontally, and by depth) and extrapolate individual vent flux estimates to a margin-wide estimate of total methane flux ($0.05 \text{ kg yr}^{-1} \text{ m}^{-2}$). Specifically, the spatially normalized results indicate that methane vents are more common on the continental shelf than in deeper waters and that the highest density of vents occurs outside the Strait of Juan de Fuca. Results further indicate that methane venting is not necessarily concentrated at the upper limit of the gas hydrate stability zone.

Although methane venting has been known for some time to be a widespread phenomenon on the Cascadia Margin in a general sense, the presented results are novel and contribute to the existing knowledge base in that they are the first attempt to compile a comprehensive database of specific methane vents on this margin. Additionally, I suspect that many of the reported vent locations were not previously specifically known to science until the authors identified them and extracted their positions from archived sonar data. Moreover, this manuscript represents the first effort of this magnitude and level of detail to estimate a value for margin-wide flux on the Cascadia Margin. Finally, the explicit discussion of the quantitative uncertainties associated with deriving flux estimates from acoustic data and extrapolating those estimates to a margin wide are particularly valuable as many researchers in this disciplinary community are concerned with how to best "scale-up" discrete observations of methane venting on continental margins to meaningful representative terms in global carbon budgets. For these reasons, I expect that the presented results will be of substantial interest to researchers working on problems of gas hydrate dynamics, subsurface methane transport processes, and large-scale carbon cycling on the Cascadia Margin as well as other continental margins around the world. Once published, I expect that these results will be widely cited by a wide range of investigators serve as background for future detailed exploration efforts focused on methane vent systems and associated chemosynthetic ecosystems on the Cascadia Margin. Accordingly, it is my opinion that the manuscript is suitable for publication in Nature Communications once the relatively minor concerns indicated below are addressed.

I find that the presented methods for deriving methane vent locations from the sourced acoustic data are technically sound. Regarding calculations of flux from acoustic data and extrapolation to total marginal flux rates, it seems that the methods are technically sound, but I was initially left wondering whether the substantial assumptions and quantitative uncertainties necessarily associated with these calculations render the final total marginal flux result so uncertain that it is meaningless, or nearly so. This is not a quibble with the work of the authors but an issue inherent in calculations of this nature. Ultimately, I believe that the authors' clear statement of assumptions and explicit handling of uncertainty is valuable for the research community in terms of identifying observational/ground truthing needs for these types of estimates and may well serve to initiate further productive research.

General Comments:

There are a few points where increased methodological detail would be valuable:

- Explicitly detail how the local upper limit of the gas hydrate stability zone was calculated or explain why the nominal value of 500m was used.

-In my experience, the intensity of the acoustic reflection from a methane flare can vary significantly based upon where the flare falls in the singlebeam/split-beam (EK60) sonar footprint. For example, a given flare that passes through the center of the sonar footprint will have a much higher intensity reflection than the same flare that passes through the outer edge of the sonar footprint. It is not clear to me how the position of the flare within the sonar footprint/insonified water volume is fully accounted for/ corrected in the method for deriving flux.

-With regard to the locations of venting (table 2), it would be useful to indicate the spatial/vertical uncertainty associated with reported coordinate and depth values. On a related point, the authors should clearly explain how "duplication of vent-counting was avoided" Many of these vents are very closely positioned. For example, was there a quantitative method for determining if two closely spaced points (perhaps within the spatial uncertainty of the observations) were individual vents or a repeat observation of a single vent?

-There are many sources of quantitative uncertainty inherent in calculations of individual vent flux and total margin flux. The authors address some of these individually at different points in the manuscript. I do not require it, but I think it would be very helpful for the authors to list every source of flux uncertainty and its relative contribution to total vent and margin flux uncertainty in a tabular format. I think this would make clear the reliability of stated flux values and be widely referred to by investigators attempting similar flux estimates for other margins.

Figures:

Figure 1:

-It is difficult to identify vent markers over the bathymetry color map (e.g. red marker over orange bathymetry).

-It is very difficult to identify the light gray lines indicating ship tracks in panel (b).

Figure 3:

-Suggest labeling top panel (a) and bottom panel (b) and referring to them as such in the caption in order to be consistent with other figures in manuscript

-Please clearly indicate what the yellow dotted line in the top panel represents and how it was generated

Specific Comments:

Line 45: I object to the use of the term "comprehensively" given that there are substantial portions of the Cascadia Margin that have not been observed by sonar. The existing sonar data from the margin were comprehensively reviewed, but the coverage of those data is far from comprehensive.

Line 58: The first letters of "Strontium" and "Lithium" should be lower case as chemical elements are not capitalized.

Line 67: Suggest inserting the word "proximity" immediately after "landward"

Line 80: Suggest replacing "Only" with "Very"

Line 184 and 187: Quick calculation with the provided values suggest that the word "radius" should be "diameter" in both cases.

Line 189: Suggest changing "amount of gas venting" to "number of gas vents"

Line 191: Reference #75 is not in the manuscript's reference section

Line 200: Suggest replacing "larger" with "greater"

Line 284 – 292: The conclusion that tides exert a strong influence on methane flow intensity for the selected data shown in fig 5 is convincing. However, this is a very short duration (less than 24 hours) example from a single vent. More supporting data is needed to reasonably conclude a tidal influence on total margin methane flux. If the data are available, I would suggest similar analysis for a number of seeps over longer periods with a statistical (transform time series to frequency space) demonstration that flow variability is linked to tidal excursion. Such analyses would strengthen assumption a) on line 345.

Line 281: I believe "low" should be "flow"

Line 315: Were the depth bins <250m, 250-1000m, and >1000m chosen in an arbitrary manner? It might make more sense to link the binning to physiographic boundaries such as the shelf break or foot of the slope.

Line 340: "define" should be "defined"

Line 351: Suggest replacing "long-wavelength" with "low-frequency"

Line 364-365: Eliminate the repetition of the word "values"

Line 439: Suggest replacing "vintage" with "historical"

Lines 453-461: Some clarity is necessary here. In lines 454-455 it is stated that "vents are mostly clustered at the two main canyons (Barkley and Juan de Fuca Canyon), which may also be simple result of cruise track density" and in lines 459-461 it is stated that "The impression that most vents occur across the shelf around the entrance to the Juan de Fuca Strait and the slope regions and canyon heads of Barkley and Juan de Fuca Canyon may be representative of the true vent distribution." These statements seem to be contradictory and the overall intent of the authors is muddled.

Lines 475-479: The authors indicate that flares interpreted to be from a single vent, based upon sonar data analyses, have been determined to be from multiple smaller vents when imaged with an ROV. How is this accounted for in the total vent count? Can this information be used to better quantify uncertainty in the total vent count?

Line 505: Suggest including a reference to support the assertion that gas escaping in shallow water on the shelf will reach the atmosphere. Also suggest qualifying "shallow water" in this context. It would be interesting to know how many of the observed vents on the Cascadia margin the authors believe are shallow enough to contribute methane to the atmosphere.

Reviewer #2 (Remarks to the Author):

This manuscript seeks to describe gas venting along the Cascadia Margin, and includes the spatial distribution and estimates of gas flux rates based on acoustic data. There are two important interpretational issues with this manuscript that caused me concern when reviewing it. Given that these concerns are related to the central issues of the manuscript, I feel that the author's need to fully address them prior to additional review. This will likely involve substantial additional work.

My first concern is related to the flow rate estimation, which is a major feature of the manuscript. The author's reference a method by Veloso et al. [ref 51 in the manuscript], and one of the basic requirements for the Veloso paper is knowledge of a bubble-size distribution. As the authors of the

present manuscript state, they don't have one. Nor is there any evidence I'm aware of that suggests that nature has provided an 'average' bubble size distribution that could be guessed at. Nor do the authors state what their guess actually is. There is a paucity of bubble-size distribution measurements, but the observations that do exist are as indicative of a lack of a consistent bubble size distribution as they are a canonical form (see, for example, Veloso et al., Figure 11 [ref 51 in the manuscript]). To be fair there seems to be a consistent overall range of bubble sizes being expelled from the seafloor, but evidence suggests there is much inconsistency throughout that range. The reason that this is important is that without a known bubble size distribution, there exists a large ambiguity between observed acoustic backscatter at a single frequency and the volume of the bubbles generating the backscatter. Consider, for example, an observation in which acoustic backscatter is observed to have a target strength of negative 20 dB. Assuming, for the sake of simplicity, that a gas bubble's scattering cross section is given by its geometric cross section, this observation could be explained by 10,000 1mm bubbles or 400 5mm bubbles, with total gas volumes that vary by a factor of 5 for the two estimates (this relationship is predicted, in fact, by Veloso et al's equation 13b). This is a large range, and for more complicated (or natural) bubble size distributions than the example provided here, lack of knowledge of the distribution can lead to even larger errors in the estimate of gas volume. These errors in bubble volume, in turn, corrupt the estimate of flux. With some additional work, the author's might be able to bound the flux estimate within a few orders of magnitude, but the estimate of flux stated in the abstract is misleading in terms of its uncertainty and, given the information on the bubble sizes available to the author's, likely to be wrong.

The second interpretational issue I have with the manuscript are the author's hints at a connection between the venting locations and the 500-m isobaths, and seeming 'support ... of venting activity at the feather edge of gas hydrate stability'. There may be a connection, but the author's have presented their data in what I would consider to be a misleading way. The authors have not provided any statistical bounds on their data (Figure 4, a, b, and c), and their histogram bin widths look exceedingly noisy and the results are very noisy. There are other local maxima – nearly as high as the 500-m isobaths – near 200 m and 950 m as well. How should we interpret these? There is real danger of misinterpretation here: a quick look at the NCEI data map server shows vessels doing a lot of cross-transects near the 500 m isobaths, which would provide an alternative explanation for the peak in the histogram. My concern is that while the authors clearly recognize some of the issues with respect to trackline and seeps observed per line km traveled, they are still presenting interpretations and data-derived products (the normalizations that are misleading. And this issue of excess venting at the edge of hydrate stability is one that has received a lot of attention with too little data.

A more minor issue is that the author's have not defined what they mean by an 'individual vent location', or explained how duplication of vent-counting was avoided (e.g., with what accuracy and precision did they locate, and subsequently count, the vents). There are established (published) methods for doing this.

Other minor technical issues include:

- Are flow rates provided at STP, or at the seabed? This should be obvious from the first mention of them in the abstract.
- I suspect the R/V Shimana might actually be the FSV Bell Shimada? I think NOAA calls these vessels Fisheries Survey Vessels, not Research Vessels.
- The Shimada and Miller Freeman likely have 11 degree 18 kHz sounders, not 7 degree 18 kHz sounders as stated in the text. I'm not aware of any 7 degree 18 kHz sounders. This might be important in terms of extrapolation of the observations to a larger area (see, for example, line 227)

Reviewer #3 (Remarks to the Author):

Are they novel and will they be of interest to others in the community and the wider field?

The data set is novel and will be of interest to others in the community as will be the discussion on the data analysis. For it to be interesting to the wider field a sounder discussion would be helpful.

If the conclusions are not original, it would be helpful if you could provide relevant references.

The major achievement of this work is the combination and thorough analysis of various extensive datasets to produce a map of vent site distributions in the area of the Cascadia margin (Fig.1) and a new methane flux budget for the area. The latter is not without difficulties despite the large data set. Next to the new map and flux numbers, the new finding is the predominant venting on the shelf compared to the 500 m isobar.

Conclusions on tidal variations are presented in Lit. 39 (Römer et al., 2016); there are not sufficient data for a discussion on climate change impacts; a publication on the acoustic data interpretation, the development of the numerical model and the importance of bubble size distribution and flux rates is also presented in Lit 51 (Velooso et al., 2015).

Is the work convincing, and if not, what further evidence would be required to strengthen the conclusions? On a more subjective note, do you feel that the paper will influence thinking in the field? Please feel free to raise any further questions and concerns about the paper.

The work on the new data set and vent locations is convincing. One could strengthen the interpretation on vent distribution, discuss and exclude a possible bias based on sampling routines, and discuss the data in relation to geological occurrences or sedimentation processes. The finding that venting is more pronounced on the shelf in terms of numbers but not flux rates could be strengthened and discussed in some more detail. As it is the discussion strongly focusses on the evaluation of the data analysis.

We would also be grateful if you could comment on the appropriateness and validity of any statistical analysis, as well the ability of a researcher to reproduce the work, given the level of detail provided.

Please see below for details

What are the major claims of the paper?

The authors present a unique dataset, revealing a new and more complex insight into the vent distribution along an active margin. They perform a robust data analysis of this enormous dataset using state of the art solutions. As a result they present a geographical map on the distribution of the vent sites and charts on the associated methane fluxes followed by a short comparison with a global budget. The strength of the paper is the data set and the new approach on the normalization of the regional data revealing a strong shift in the vent distribution compared to earlier approaches. The authors lead on open discussion on the matter of methane flux budgeting using acoustic data.

I would state two concerns: One is based on the nature of the data set the other is based on the discussion of the data:

Even so the data set is large the budgeting of methane flux is difficult - if not impossible. The authors state an error of "at least 20%", which is misleading as this error is caused by the assumptions on gas

bubble distribution only. There are further errors such as data quality and sampling bias. Average methane fluxes have a variance of $\pm 100\%$. The regional budget varies by two orders of magnitude; however, it is not clear what this variation is based upon. While the vent site distribution is a great achievement, the calculated flux budgets might have to be treated with more care – even so the presented data set is by far the best there is so far.

Line 191 – 197 state that the authors will - at this point - restrain from the assumption of randomly distributed vent occurrences. Later on there is an areal estimate based on the normalization using bathymetric depth distribution even so it is also argued that, e.g., most of the 500 m vent sites cluster at Clayoquite canyons.

There is little discussion/interpretation on the vent distribution, which I feel is the strength of this paper. There is no relation to a geological map, no conclusions drawn in comparison with further canyons, river outflows or other geographical and geological features. The comprehensive data set might allow this. Instead a tidal relation and climate change issues are in focus of the discussions, both of which do not seem to be the strength of the data set and have been discussed elsewhere.

More detailed comments regarding the data set and consequent budgeting /Normalization

Do these features of the data set cause a bias in methane budgeting?

The shelf: The coverage on the shelf, especially the upper 100 m, seems extremely limited (Line 229ff; Figure 3d). Is this area included? Unlike deeper areas, the shelf was investigated in a structured pattern with less emphasis on vent areas. How does this affect the data? In line 189 it states: the amount of gas venting is likely vastly underestimated (regarding small foot print on the shelf) but this is not been taken up again later in the budget discussion.

In the Appendix (Line 651) it is stated that “Overlap between different track lines either from the same or other cruises in different years was included in the integration of all footprint area values”. It is not quite clear to me if the overlaps are added up – unlike the vent sites – or if they are canceled out. In case the overlap is added up, the different approaches regarding the surveys might be of some importance: On the shelf the surveys rarely cross an area twice while surveys at larger depth are much more site specific with a much larger overlap.

Is there an impact on the budget caused by the dense sampling at Clayoquite Slope? It seems that only 70 flow estimates originate from depths > 250 m. At Clayoquite 114 estimates have been detected some of which are repetitions. Nevertheless, the number might be high in comparison.

Line 177 ff The majority of recordings collected in the NOAA database originate from depth of < 750 m. If I am not mistaken about 800 vent sites originate from the evaluation of this data set (Line 111-113: 1030 vents, out of which 182 have been published). Does this affect your vent distribution map?

Is a bias introduced into your evaluation due to large areas not having been sampled at all? Large areas below ~ 1200 m (45° - 48° N) do not have a single track line crossing them. Especially in scenario A (bathymetric depth distribution betw. coast line to deformation front) of the normalization this is likely to introduce a significant shift? Is the scenario used for the regional methane flux budgeting?

Line 282ff, Line 316 ff & Fig. 7c: natural or artificial flow rate variations of 2 – 18 L/min induce errors much larger than 20%. How does this finding affect your calculation? In Line 316ff average flow rates have uncertainties as large as the flow rates themselves. How does this effect your calculations?

Even so the data originate from 10 years it can be assumed that each site was only visited once with few exceptions and that most of the data originate from the same season (current regime). Especially on the shelf and upper slope this might be of importance (upwelling; current velocities and volume).

Comparison to other data

For the comparison the authors might want to point out that these fluxes originate from acoustic measurements, thus, exclude dissolved methane. Bubbles can also include other gases such as N₂ or O₂, which exchange with CH₄ during the ascent. Clark et al. (2000) postulate that at Coal Oil Point 25-60% of methane gas dissolves during ascent through water column. Is there any assumptions being made in the calculations?

Line 385: 0.2 and 27% - I might have missed some information but it's not quite clear to me how the authors got to these numbers? The lowest number of the global budget is as high as their highest.

Lines 382: It seems that the flux is based on the bathymetric depth distribution instead of the footprint area normalized data? Is this correct? Earlier on the authors showed a significant shift between scenarios.

Comments on Discussion

Line 454: May it be worthwhile, to add a map including a more detailed bathymetry and/or tectonic features to support discussion on geological drivers of venting, which might be supported by the new dataset. Are there any more canyons in the area? Torres et al., 2009 discuss venting on the shelf as a result of collision of a buried ridge and consequent uplift, yet another local geological trigger? Why are there more vent sites on the shelf but fluxes low? In Fig. 1 there seems to be another cluster of vents on the shelf close to outflow of major rivers: Willapa, Chehalis and Columbia. Could this support any hypothesis (Organics and high sediment flux) or might it be due to a sampling artefact?

Line 440 ff: Did you include all these data in your evaluation or is this only a general list? Before and after you refer to the local data.

Line 477: may this be part of the reason for higher flow rates assigned to the deeper vent sites? How is this tackled in the calculations?

Line 482: why should there be more bubbles only because the flare does not have a single source?

Line 494: the gas hydrate dissociation would not necessarily follow the 500m isobaths but be related to ocean currents and their changes. They determine the temperature distribution. The area is known to encounter upwelling. Still I agree, it would not be locally confined; even so currents in canyons and at the outflow of the JdF St could be specific.

Line 503 ff: there are publication for the area, e.g. from the Coal Oil Point (Mau et al, 2007) or onshore Hydrate Ridge (Rehder et al., 2002). Overall the short discussion seems rather superficial and has little overlap with the dataset. You might want to consider taking this out.

Line 508 ff: This list partly seems to be a repetition of things just said. Number 5 is what fits best to your data set. Number 1 is tackled in Römer et al., (2016) and other literature; Numbers 2 – 4, I would attach to the data evaluation while for #5 the new data set could be very valuable.

Especially on the shelf and upper slope seasonal changes might play a significant role (upwelling; undercurrent; river discharge...)

Abstract

You might want to add a range/error to the methane flux data.

Introduction

The first paragraph is incoherent (e.g. active, passive, Arctic margin is a mix) and lists all “hot” topics vaguely related to methane fluxes. Less might be more and would allow being more precise. E.g., the CH₄ outflow has a rather very limited impact on the carbon in marine sediments, “Natural flux of gases” (37) is a very vague expression.

Line 78: natural flux of methane: out of the sediment or quantifications within the sediment

Line 89-94: The text refers to data are on pore water measurements of CH₄ fluxes (dissolved and /or within the sediment?). This might be redundant to this data set.

Literature: It might be worthwhile to mention Coal Oil Point, which is a well investigated vent site on the shelf (California) in terms of methane transport and fluxes (e.g. Clark et al., 2000; Mau et al., 2007). Further, there is a publication by Rehder et al., 2002 regarding ocean-atmosphere gas fluxes of methane at Hydrate Ridge and the shelf area. McGinnis et al. (2006) – Black Sea; Westbrook et al. (2009) Svalbard (e.g.)

Methods

Without any information on the methods I find it very hard to read or assess the results and Fig. 3.

Appendix Line: 571: The temperatures are higher : < 500 m: 5-9°C; 500 – 1000 m: 3.5 – 5°C

Results

Fig. 1 There will be distinctions between the shelf, 250 -1000 m, and >1000 m. Apart from the 500 m isobar these depth ranges are not resolved in the Figure. In the right panel one could leave out the vent sites to better visualize the tracks and change the gray lines to a slightly darker gray.

Fig. 3 The figure caption is rather short. It might just be me but I had trouble understanding Fig. 3 at his point and thus had difficulties following the normalization. Percentages are really difficult to follow up.

Line 203-204: with the very low number of lines crossing at the DF - apart from the area offshore Vancouver Island - I would restrain from this statement

Acoustic quantification of gas emissions

Bubbles below 500 m might have a gas hydrate coating (e.g. Rehder et al., Sauter et al.). The coating should alter the response to soundwaves compared to bubbles without. Is this difference included?

Does your model include the different bubbles size distribution encountered at vent sites in different depth ranges?

Line 282ff: significant variations in the flow rate estimate.. may represent real changes.. but could also be a result due to noise in the data. This is a statement that completely undermines your calculations. What's the error introduced by the quality of the data?

Fig. 5 The comparison between the flow rates and the tidal cycle is not very clear – you might want to add another cycle. (Römer et al., 2016)

How about the currents which are much stronger at depth < 500 m? The dissolution should be faster as should be the bubble dispersion. This is taken into account in your calculations, is it not?

Margin-wide gas flux

Line 352 This assumption might hold some difficulties: the clustering along the canyons might not fit the overall “physical constraint of the geographical region”. The idea of rise rates being the same all over the shelf is a daring thesis as, e.g., the current regimes and river discharges vary considerably in this geographical region.

Line 357: It seems like 1600 m should be your maximum depth since the data set is too scarce below this depth.

Line 359: Is the tidal forcing included in your flux calculations?

Lines 369 ff – what defines minimum and maximum flow rates where before there was not such a thing? For the first time there is a 2 order of magnitude variation for your rates.

Reply to individual review comments

Distributed natural gas venting offshore along the Cascadia Margin

M. Riedel, M. Scherwath, M. Römer, M. Veloso, M. Heesemann, and G.D. Spence

We would like to thank all reviewers for their detailed comments provided which were very useful and guided us through additional work. Using the many (often similar) comments we were able to considerably improve the presented work. The main advances made are:

- Definition of bubble size distribution and bubble-rise rates for Cascadia using video footage from Ocean Networks Canada (ONC): Calibrated videos were used to trace individual bubbles to calculate bubble-size dependent rise-rates. Also, acoustic data from times when vessel was stationary on top of a gas plume were used to get average rise-rates for the bubble streams. These data do result in very similar overall values, compared to the known literature.
- Depth-dependent bottom water temperature definition: Although our data span more than 15 years we used average bottom-water temperatures from international data bases for the region off Cascadia and defined a best-fit polynomial to the spread in observations (regionally and in time). This is a better approximation than a constant value used before; yet, it may not catch all seasonal or decadal variations. However, it reduces the overall uncertainty per individual calculation.
- We introduced a new tidal-modulation model on gas emission and compare these new values to a pure linear extrapolation of the initial flow-rate estimates. Using the long-term observations (previously published by our team) and using similar observations made at gas seeps world-wide, a tidal-forcing model is used to first predict gas flow rate of a single flare-site per tidal cycle, which is then extrapolated to annual gas emission values.
- Normalization of our gas seepage sites by the foot-print of the various acoustic echo-sounder types were updated based on the revised aperture-values and re-applied. As previously seen, the footprint is low across the shallow shelf and normalization by footprint area emphasizes this region relative to deeper water depths.
- We removed detailed discussion on the issue of the 500 m isobath and links to gas hydrate dissociation as it muddles the overall intent of our contribution.

Many additional comments were provided by the three reviewers which we address in detail below. However, some comments were identical/similar between the individual reviewers and we then abbreviate our replies and refer back to the initial reply (and this introduction).

Overall, we have significantly updated the manuscript and provide a version using track-changes. As this may be rather difficult to read, we also include a clean-copy with all changes applied. Please note, for the ease of reading and understanding the impact of different techniques employed, we created two additional tables.

With kind regards,

Michael Riedel on behalf of all co-authors

Detailed reply to comments made by Reviewer #1

R1.1 I find that the presented methods for deriving methane vent locations from the sourced acoustic data are technically sound. Regarding calculations of flux from acoustic data and extrapolation to total marginal flux rates, it seems that the methods are technically sound, but I was initially left wondering whether the substantial assumptions and quantitative uncertainties necessarily associated with these calculations render the finally total marginal flux result so uncertain that it is meaningless, or nearly so. This is not a quibble with the work of the authors but an issue inherent in calculations of this nature. Ultimately, I believe that the authors' clear statement of assumptions and explicit handling of uncertainty is valuable for the research community in terms of identifying observational/ground truthing needs for these types of estimates and may well serve to initiate further productive research.

Reply: The reviewer is correct that many assumptions and extrapolations have to be used in order to achieve a margin-wide flux. However, with the new constraints from video-observations and data we render our calculations as “currently best possible estimates” at each individual location. As data span over 15 years there are certainly many factors which we cannot catch in this margin-wide flux estimate (e.g. decadal variations in oceanography settings). Also, the data we use are certainly point-data and questions immediately arise whether they are representative for the entire region and all water depths investigated.

However, we made some additional effort to define the overall uncertainty in the estimates and carry out various calculations to get a possible spread in the final flux rate. Perhaps surprisingly, these values do not differ too much from each other (see Table 4). Also, our work highlights several areas where improvements in the input variables may be most significant and can thus guide new field work.

R1.2 There are a few points where increased methodological detail would be valuable: Explicitly detail how the local upper limit of the gas hydrate stability zone was calculated or explain why the nominal value of 500m was used.

Reply: The value of 500 m was taken from the literature of gas venting across Cascadia (e.g. Hautala et al., 2014); we only use it to place our results into this overall context and ongoing debate within the research community. We have not tested any new gas hydrate stability modeling to re-define this depth limit. As a result of this comment (and those made by the other reviewers) we down-scaled our discussion of this point and removed it from the abstract.

R1.3 In my experience, the intensity of the acoustic reflection from a methane flare can vary significantly based upon where the flare falls in the singlebeam/split-beam (EK60) sonar footprint. For example, a given flare that passes through the center of the sonar footprint will have a much higher intensity reflection than the same flare that passes through the outer edge of the sonar footprint. It is not clear to me how the position of the flare within the sonar

footprint/insonified water volume is fully accounted for/ corrected in the method for deriving flux.

Reply: First, we use data where we can trace the bubble stream within the 3D-cone of illuminated acoustic ranges to the seafloor and do not use flares that are cut off (either in depth or side-ways). The entire range of acoustic data acquisition parameters (tilt of ship, pitch, roll) are also included in the calculation. Details to this matter can also be found in the original paper on this method by Veloso et al. (2015).

R1.4 With regard to the locations of venting (table 2), it would be useful to indicate the spatial/vertical uncertainty associated with reported coordinate and depth values.

Reply: We cite in Table 2 the source of acoustic data (e.g. EK60, EM302) and together with the water depth (which is certain to within 1% of the water depth for the echosounders used) the horizontal uncertainty can be defined by (a) footprint of the system or (b) the geometrical spread of the bubble-plume imaged. We always use the centre of the acoustic flare image at the seafloor for the lat/long values reported in Table 2. Horizontal uncertainties are uniformly defined as ½ the footprint size for the EK60 systems. For EM systems, a beam-width angle of 4° was used to define the footprint uncertainty at the seafloor. ROV-based systems are very precise but horizontal uncertainty of the ROV relative to the vessel is defined by the USBL system, which is accurate to within 1% of the water depth. We added a column to Table 2 for this horizontal uncertainty value.

R1.5 On a related point, the authors should clearly explain how “duplication of vent-counting was avoided” Many of these vents are very closely positioned. For example, was there a quantitative method for determining if two closely spaced points (perhaps within the spatial uncertainty of the observations) were individual vents or a repeat observation of a single vent?

Reply: The measure to distinguish such technique is the footprint of the acoustic data. If a new flare is detected at a distance to an existing flare that is smaller than the full footprint of the EK60 data used to find the flare, then the flare is deemed a duplicate. Otherwise, if the distance is larger than the footprint, it is counted as a new independent flare location. For EM-based systems, the detection threshold is defined by the beam-angle sub-division width (4°) as described above. Text explaining this was added to the revised manuscript.

R1.6 There are many sources of quantitative uncertainty inherent in calculations of individual vent flux and total margin flux. The authors address some of these individually at different points in the manuscript. I do not require it, but I think it would be very helpful for the authors to list every source of flux uncertainty and its relative contribution to total vent and margin flux uncertainty in a tabular format. I think this would make clear the reliability of stated flux values and be widely referred to by investigators attempting similar flux estimates for other margins.

Reply: We have augmented the text to include a more complete discussion on uncertainties including those from assumptions (e.g. BSD, rise-rate, clean vs. dirty) and derive a better constrained “total uncertainty” to the reported average values.

R1.7 Figure 1: It is difficult to identify vent markers over the bathymetry color map (e.g. red marker over orange bathymetry). It is very difficult to identify the light gray lines indicating ship tracks in panel (b).

Reply: We have modified Figure 1 accordingly.

R1.8 Figure 3: Suggest labeling top panel (a) and bottom panel (b) and referring to them as such in the caption in order to be consistent with other figures in manuscript. Please clearly indicate what the yellow dotted line in the top panel represents and how it was generated

Reply: We assume that the reviewer was referring to Figure 5: We changed to (a) and (b) and removed the yellow line.

R1.9 Line 45: I object to the use of the term “comprehensively” given that there are substantial portions of the Cascadia Margin that have not been observed by sonar. The existing sonar data from the margin were comprehensively reviewed, but the coverage of those data is far from comprehensive.

Reply: changed according to suggestion.

R1.10 Line 58: The first letters of “Strontium” and “Lithium” should be lower case as chemical elements are not capitalized.

Reply: changed according to suggestion.

R1.11 Line 67: Suggest inserting the word “proximity” immediately after “landward”

Reply: changed according to suggestion.

R1.12 Line 80: Suggest replacing “Only” with “Very”

Reply: changed according to suggestion.

R1.13 Line 184 and 187: Quick calculation with the provided values suggest that the word “radius” should be “diameter” in both cases.

Reply: changed according to suggestion.

R1.14 Line 189: Suggest changing “amount of gas venting” to “number of gas vents”

Reply: changed according to suggestion.

R1.15 Line 191: Reference #75 is not in the manuscript's reference section

Reply: references and sub-sentence was removed

R1.16 Line 200: Suggest replacing “larger” with “greater”

Reply: changed according to suggestion.

R1.17 Line 284 – 292: The conclusion that tides exert a strong influence on methane flow intensity for the selected data shown in fig 5 is convincing. However, this is a very short duration (less than 24 hours) example from a single vent.

Reply: Yes, the reviewer is correct; however, we do not have any longer duration acoustic EK60 survey to show such close relationship. We refer to Römer et al. (2016) for long-term (13 months) observations in the manuscript.

R1.18 More supporting data is needed to reasonably conclude a tidal influence on total margin methane flux. If the data are available, I would suggest similar analysis for a number of seeps over longer periods with a statistical (transform time series to frequency space) demonstration that flow variability is linked to tidal excursion. Such analyses would strengthen assumption a) on line 345.

Reply: Based on the data reported by Römer et al. (2016) we have developed a tidal-forcing model to estimate gas flow over a single tidal cycle. We stacked data over 40 tidal cycles of varying intensity, normalized the response and the integral of this function over 12 hours defines the flow per tidal cycle. See new Methods section for details.

R1.19 Line 281: I believe “low” should be “flow”

Reply: changed according to suggestion.

R1.20 Line 315: Were the depth bins <250m, 250-1000m, and >1000m chosen in an arbitrary manner? It might make more sense to link the binning to physiographic boundaries such as the shelf break or foot of the slope.

Reply: We defined those boundaries reflecting the physiography of the entire margin (shelf break extent varies from North to South) and the spread of our acoustic data. The water depth was used to calculate slope angle. As the shelf-break is defined as the location of significant change in slope angle we used this as guide for the depth-extent of the shelf. The image to the left shows a map of the slope angle (in °) along the central portion of the Cascadia margin combined with the 250 isobath (thick black line). With the exception of glacial-channels on the shelf at the entrance/exit of the Juan de Fuca Strait, the 250m isobaths is along the region of first significant change in slope angle used to define the extent of the shelf in general terms. Therefore, the 250 m

water depth cutoff is a meaningful measure for the shelf extent along the Cascadia margin.

The cutoff at the 1000m depth was used due to the observations in gas flares (see Figure 7), where a “gap” exists after 1000 m water depth and deeper flares resulted in significantly higher average flow rate estimates. As only 23 flares were observed in the depth range between 1000 m and 1200 m the change in extrapolated (and normalized) gas flow rate for Cascadia is relatively small (< 10%) and thus within the overall uncertainty defined from all sources.

We modified the manuscript for clarification and added new text to the Method section and discussion. We also submit a supplementary map showing the 250 m isobath and slope-angle map for the entire Cascadia margin.

R1.21 Line 340: “define” should be “defined”

Reply: changed according to suggestion.

R1.22 Line 351: Suggest replacing “long-wavelength” with “low-frequency”

Reply: changed according to suggestion.

R1.23 Line 364-365: Eliminate the repetition of the word “values”

Reply: changed according to suggestion.

R1.24 Line 439: Suggest replacing “vintage” with “historical”

Reply: changed according to suggestion.

R1.25 Lines 453-461: Some clarity is necessary here. In lines 454-455 it is stated that “ vents are mostly clustered at the two main canyons (Barkley and Juan de Fuca Canyon), which may also be simple result of cruise track density” and in lines 459-461 it is stated that “The impression that most vents occur across the shelf around the entrance to the Juan de Fuca Strait and the slope regions and canyon heads of Barkley and Juan de Fuca Canyon may be representative of the true vent distribution.” These statements seem to be contradictory and the overall intent of the authors is muddled.

Reply: These two comments not necessarily contradict each other. On the one hand, line density is higher at the entrances of the JdF Strait as many cruises originate there and thus more vents could be found as we just looked more often. Otherwise, the nutrient-rich outflow may indeed favor methane gas production by microbes and thus gas venting. We unfortunately lack data to fully address the origin of the observation. We draw no further conclusions from this “fact” in the gas flare observations; thus we do not see any reason for any major change. The text was overall adjusted to reflect this point and comment.

R1.26a Lines 475-479: The authors indicate that flares interpreted to be from a single vent, based upon sonar data analyses, have been determined to be from multiple smaller vents when imaged with an ROV. How is this accounted for in the total vent count?

Reply: In order to reduce confusion between terms ‘vent’ and ‘flare’, we refer to only ‘flares’. A vent is a term that often includes numerous gas outlets, carbonate concretions and bacterial mats, thus can be a wide-spread region affected by gas venting. ROV-based observations of individual gas flares only constitute a new “flare count” if no acoustic data have covered the region.

R1.26b Can this information be used to better quantify uncertainty in the total vent count?

Reply: Unfortunately, no. However, video footage at single flares can be used to define bubble-radius, etc., and thus help in the quantification and thus indirectly reduce uncertainty.

R1.27 Line 505: Suggest including a reference to support the assertion that gas escaping in shallow water on the shelf will reach the atmosphere. Also suggest qualifying “shallow water” in this context. It would be interesting to know how many of the observed vents on the Cascadia margin the authors believe are shallow enough to contribute methane to the atmosphere.

Reply: We restrain from making any linkages to possible gas reaching the atmosphere. This decision was also made in reply to a different comment made by another reviewer. The portion of text was removed from the revised manuscript.

Detailed reply to comments made by Reviewer #2:

R2.1 This manuscript seeks to describe gas venting along the Cascadia Margin, and includes the spatial distribution and estimates of gas flux rates based on acoustic data. There are two important interpretational issues with this manuscript that caused me concern when reviewing it. Given that these concerns are related to the central issues of the manuscript, I feel that the author's need to fully address them prior to additional review. This will likely involve substantial additional work.

Reply: We thank the reviewer for the careful review and suggestions. Substantial new work was conducted and included to address the comments made.

R2.2 My first concern is related to the flow rate estimation, which is a major feature of the manuscript. The author's reference a method by Veloso et al. [ref 51 in the manuscript], and one of the basic requirements for the Veloso paper is knowledge of a bubble-size distribution. As the authors of the present manuscript state, they don't have one. Nor is there any evidence I'm aware of that suggests that nature has provided an 'average' bubble size distribution that could be guessed at. Nor do the authors state what their guess actually is. There is a paucity of bubble-size distribution measurements, but the observations that do exist are as indicative of a lack of a consistent bubble size distribution as they are a canonical form (see, for example, Veloso et al., Figure 11 [ref 51 in the manuscript]). To be fair there seems to be a consistent overall range of bubble sizes being expelled from the seafloor, but evidence suggests there is much inconsistency throughout that range. The reason that this is important is that without a known bubble size distribution, there exists a large ambiguity between observed acoustic backscatter at a single frequency and the volume of the bubbles generating the backscatter. Consider, for example, an observation in which acoustic backscatter is observed to have a target strength of negative 20 dB. Assuming, for the sake of simplicity, that a gas bubble's scattering cross section is given by its geometric cross section, this observation could be explained by 10,000 1mm bubbles or 400 5mm bubbles, with total gas volumes that vary by a factor of 5 for the two estimates (this relationship is predicted, in fact, by Veloso et al's equation 13b). This is a large range, and for more complicated (or natural) bubble size distributions than the example provided here, lack of knowledge of the distribution can lead to even larger errors in the estimate of gas volume. These errors in bubble volume, in turn, corrupt the estimate of flux. With some additional work, the author's might be able to bound the flux estimate within a few orders of magnitude, but the estimate of flux stated in the abstract is misleading in terms of its uncertainty and, given the information on the bubble sizes available to the author's, likely to be wrong.

Reply: We thank the reviewer for this detailed comment and analysis. With the help of Ocean Networks Canada (ONC) we secured digital video imagery of ROV-based observations of gas venting. The videos were used to (a) define a range in bubble size (new Figure 9a) and rise-rates (Figure 9b-d). While video-observations are sparse, we used acoustically defined rise-rates to compare our results from the ONC data. The observed range in rise-rates across numerous vents suggests a rather "uniform" behavior of bubbles emitted at these vent-sites. This is used as an indicator to apply the bubble-size distribution for the entire margin. The use of different bubble-size distributions of course shifts the flow rate estimate; if more large bubbles exist, the flow estimate is higher (speaking in generalized terms).

R2.3 The second interpretational issue I have with the manuscript are the author's hints at a connection between the venting locations and the 500-m isobath, and seeming 'support ... of venting activity at the feather edge of gas hydrate stability'. There may be a connection, but the author's have presented their data in what I would consider to be a misleading way.

Reply: As mentioned in the general introduction, we down-scaled the discussion on the 500 m isobath and links to gas hydrate dissociation. See also reply to comment R1.2.

R2.4 The authors have not provided any statistical bounds on their data (Figure 4, a, b, and c), and their histogram bin widths look exceedingly noisy and the results are very noisy.

Reply: The data shown in Figure 4 are the water-depth distributions of gas flare occurrence and their normalizations with respect to bathymetry and footprint. Statistical bounds are the shown extent of values measured (Figure 3) and yes, the data are noisy in that not the entire margin is covered and the various data sets have different spatial coverage. Unfortunately, we cannot do much about this at this point. Maybe, future work after more cruises have been completed could be conducted.

R2.5 There are other local maxima – nearly as high as the 500-m isobaths – near 200 m and 950 m as well. How should we interpret these?

Reply: we suggest to restrain from any additional interpretation, especially since they fade with sonar-footprint normalization.

R2.6 There is real danger of misinterpretation here: a quick look at the NCEI data map server shows vessels doing a lot of cross-transects near the 500 m isobaths, which would provide an alternative explanation for the peak in the histogram. My concern is that while they authors clearly recognize some of the issues with respect to trackline and seeps observed per line km traveled, they are still presenting interpretations and data-derived produces (the normalizations that are misleading. And this issue of excess venting at the edge of hydrate stability is one that has received a lot of attention with too little data.

Reply: Yes, the reviewer is correct. The 500 m isobaths-line was a target, thus can create a bias. We reduced our overall discussion of this point in the revised manuscript.

R2.7 A more minor issue is that the author's have not defined what they mean by an 'individual vent location', or explained how duplication of vent-counting was avoided (e.g., with what accuracy and precision did they locate, and subsequently count, the vents). There are established (published) methods for doing this.

Reply: We added some text to this matter (also in reply to comment R1.5) and added a column in Table 2 to define the horizontal uncertainty in vent-location.

R2.8 Are flow rates provided at STP, or at the seabed? This should be obvious from the first mention of them in the abstract.

Reply: The values are reported not at STP but as in situ values. A depth-dependent density of methane in seawater was used to convert the flow rates in liters/min to rates in kg/min (and then extrapolated to kg/year). Where we introduce and define our two different kinds of flow rates, we specifically say “in situ.”

R2.9 I suspect the R/V Shimana might actually be the FSV Bell Shimada? I think NOAA calls these vessels Fisheries Survey Vessels, not Research Vessels.

Reply: Correct, we apologize for this error and typo.

R2.10 The Shimada and Miller Freeman likely have 11 degree 18 kHz sounders, not 7 degree 18 kHz sounders as stated in the text. I'm not aware of any 7 degree 18 kHz sounders. This might be important in terms of extrapolation of the observations to a larger area (see, for example, line 227)

Reply: Correct, we adjusted all values in Table 1, re-calculated the foot-print and modified all related analyses.

Detailed reply to comments made by Reviewer #3:

R3.1 The data set is novel and will be of interest to others in the community as will be the discussion on the data analysis. For it to be interesting to the wider field a sounder discussion would be helpful.

Reply: We augmented the discussion and also introduced new methodology to augment the impact of the manuscript.

R3.2 The major achievement of this work is the combination and thorough analysis of various extensive datasets to produce a map of vent site distributions in the area of the Cascadia margin (Fig.1) and a new methane flux budget for the area. The latter is not without difficulties despite the large data set. Next to the new map and flux numbers, the new finding is the predominant venting on the shelf compared to the 500 m isobar. Conclusions on tidal variations are presented in Lit. 39 (Römer et al., 2016); there are not sufficient data for a discussion on climate change impacts; a publication on the acoustic data interpretation, the development of the numerical model and the importance of bubble size distribution and flux rates is also presented in Lit 51 (Veloso et al., 2015).

Reply: With respect to the comment on climate change impact: yes, we have little to no data to demonstrate this. We removed a section in the manuscript that mentioned gas flux to the atmosphere.

R3.3 On a more subjective note, do you feel that the paper will influence thinking in the field? The work on the new data set and vent locations is convincing. One could strengthen the interpretation on vent distribution, discuss and exclude a possible bias based on sampling routines, and discuss the data in relation to geological occurrences or sedimentation processes. The finding that venting is more pronounced on the shelf in terms of numbers but not flux rates could be strengthened and discussed in some more detail. As it is the discussion strongly focusses on the evaluation of the data analysis.

Reply: We overall modified the manuscript and included new methods and strengthened the discussion.

R3.4 The authors lead on open discussion on the matter of methane flux budgeting using acoustic data. I would state two concerns: One is based on the nature of the data set the other is based on the discussion of the data: Even so the data set is large the budgeting of methane flux is difficult - if not impossible. The authors state an error of “at least 20%”, which is misleading as this error is caused by the assumptions on gas bubble distribution only. There are further errors such as data quality and sampling bias. Average methane fluxes have a variance of $\pm 100\%$. The regional budget varies by two orders of magnitude; however, it is not clear what this variation is based

upon. While the vent site distribution is a great achievement, the calculated flux budgets might have to be treated with more care – even so the presented data set is by far the best there is so far.

Reply: Based on the comments made by this and the other reviewers, we added a discussion on the errors and assumptions, augmented the methodology and defined an uncertainty limit.

R3.5 Line 191 – 197 state that the authors will - at this point - restrain from the assumption of randomly distributed vent occurrences. Later on there is an areal estimate based on the normalization using bathymetric depth distribution even so it is also argued that, e.g., most of the 500 m vent sites cluster at Clayoquot canyons. There is little discussion/interpretation on the vent distribution, which I feel is the strength of this paper. There is no relation to a geological map, no conclusions drawn in comparison with further canyons, river outflows or other geographical and geological features. The comprehensive data set might allow this. Instead a tidal relation and climate change issues are in focus of the discussions, both of which do not seem to be the strength of the data set and have been discussed elsewhere.

Reply: The authors felt after discussions that it would be best to restrain from too detailed interpretation of the vent distribution and linkages to the occurrence of canyons and river outflow sites. As there may well be a physical (biological?) association, it warrants further research into the existing literature and maybe conducting different cruises. All this would diverge from the main intent of our contribution.

R3.6a The shelf: The coverage on the shelf, especially the upper 100 m, seems extremely limited (Line 229ff; Figure 3d). Is this area included?

Reply: The shallowest finding is in 40 m water depth. Data shallower than this are difficult to interpret for gas flare occurrence. Statistically that was the cutoff of all normalizations.

R3.6b Unlike deeper areas, the shelf was investigated in a structured pattern with less emphasis on vent areas. How does this affect the data? In line 189 it states: the amount of gas venting is likely vastly underestimated (regarding small foot print on the shelf) but this is not been taken up again later in the budget discussion.

Reply: When calculating the amount of gas venting across the margin, we do highlight the shelf-contribution. Depending on the type of normalization, the results vary strongly. Likely, more work is warranted targeting the shallow water depths and gas flares on the shelf. However, such data do not exist yet to further reduce uncertainty in this matter.

R3.6c In the Appendix (Line 651) it is stated that “Overlap between different track lines either from the same or other cruises in different years was included in the integration of all footprint area values”. It is not quite clear to me if the overlaps are added up – unlike the vent sites – or if they are canceled out. In case the overlap is added up, the different approaches regarding the surveys might be of some importance: On the shelf the surveys rarely cross an area twice while surveys at larger depth are much more site specific with a much larger overlap.

Reply: In accordance to other reviewers' comments, we inserted text to explain how vent-duplication was avoided and Table 2 now includes a measure for horizontal location uncertainty.

R3.6d Is there an impact on the budget caused by the dense sampling at Clayoquite Slope? It seems that only 70 flow estimates originate from depths > 250 m. At Clayoquite 114 estimates have been detected some of which are repetitions. Nevertheless, the number might be high in comparison.

Reply: Clayoquite is the site of most estimates, as well as vent sites in the near vicinity (e.g. Spinnaker, Amnesiac, etc.). The values do contribute in a way of providing an average values for deep-water flares, but no further skew is produced by such duplication in location.

R3.6e Line 177 ff The majority of recordings collected in the NOAA database originate from depth of < 750 m. If I am not mistaken about 800 vent sites originate from the evaluation of this data set (Line 111-113: 1030 vents, out of which 182 have been published). Does this affect your vent distribution map?

Reply: No, the map in itself is not affected.

R3.6f Is a bias introduced into your evaluation due to large areas not having been sampled at all? Large areas below ~ 1200 m (45°-48°N) do not have a single track line crossing them. Especially in scenario A (bathymetric depth distribution betw. coast line to deformation front) of the normalization this is likely to introduce a significant shift?

Reply: The fact that certain regions have not been mapped does post a level of uncertainty in the vent counts. More cruises with more observations in the future will help to close the gap. At this point, we only consider those regions as “not sampled” (Map in Figure 8) and thus those regions do not contribute to the normalization by foot-print.

R3.6g Is is the scenario used for the regional methane flux budgeting?

Reply: Yes, when considering the normalizations, such data gaps do matter and affect the outcome.

R3.6h Line 282ff, Line 316 ff & Fig. 7c: natural or artificial flow rate variations of 2 – 18 L/min induce errors much larger than 20%. How does this finding affect your calculation? In Line 316ff average flow rates have uncertainties as large as the flow rates themselves. How does this effect your calculations?

Reply: We updated the error and uncertainty discussion and state that the 20% error only comes from assumptions on variables in the flow-rate calculation. Much higher errors are introduced when extrapolating these values.

R3.6i Even so the data originate from 10 years it can be assumed that each site was only visited once with few exceptions and that most of the data originate from the same season (current regime). Especially on the shelf and upper slope this might be of importance (upwelling; current velocities and volume).

Reply: Yes, most data are from summer months, thus we cannot really discuss longer-term or even seasonal effects. It is one source of uncertainty, but of an unknown magnitude.

R3.7a For the comparison the authors might want to point out that these fluxes originate from acoustic measurements, thus, exclude dissolved methane. Bubbles can also include other gases such as N₂ or O₂, which exchange with CH₄ during the ascent. Clark et al. (2000) postulate that at Coal Oil Point 25-60% of methane gas dissolves during ascent through water column. Is there any assumptions being made in the calculations?

Reply: We have modified the manuscript to stress that we only consider acoustically resolved vents, and now call them flares through the manuscript. Furthermore, we only considered a maximum height of 10 m of acoustic data for the flow-rate estimation and across this water depth we ignore any gas exchange.

R3.7b Line 385: 0.2 and 27% - I might have missed some information but it's not quite clear to me how the authors got to these numbers? The lowest number of the global budget is as high as their highest.

Reply: We changed the phrase and made it more clear that our (updated) average methane flow rate is divided this by the two extreme values published by Ruppel and Kessler, 2017.

R3.7c Lines 382: It seems that the flux is based on the bathymetric depth distribution instead of the footprint area normalized data? Is this correct? Earlier on the authors showed a significant shift between scenarios.

Reply: To clarify the discussion and results of using different normalizations, we added Table 4 with all results.

R3.8 Comments on Discussion

Line 454: May it be worthwhile, to add a map including a more detailed bathymetry and/or tectonic features to support discussion on geological drivers of venting, which might be supported by the new dataset. Are there any more canyons in the area? Torres et al., 2009 discuss venting on the shelf as a result of collision of a buried ridge and consequent uplift, yet another local geological trigger? Why are there more vent sites on the shelf but fluxes low? In Fig. 1 there seems to be another cluster of vents on the shelf close to outflow of major rivers: Willapa, Chehalis and Columbia. Could this support any hypothesis (Organics and high sediment flux) or might it be due to a sampling artefact?

Reply: The reviewer adds many new questions and voices interesting ideas, all warranted in general, but significantly augmenting the scope of our manuscript. We restrain from too many additional discussions on the actual vent distribution with respect to canyon sites or other tectonic features.

R3.9 Line 440 ff: Did you include all these data in your evaluation or is this only a general list? Before and after you refer to the local data.

Reply: We describe this in detail in our Results section, before the Discussion here. Historical data are used as information (lat/long/depth), and the individual locations are counted and included in the margin-wide normalized extrapolations. However, flow estimates are not made at these locations.

R3.10 Line 477: may this be part of the reason for higher flow rates assigned to the deeper vent sites? How is this tackled in the calculations?

Reply: We believe this is not the cause for higher flow rates at deeper sites. Our approach just takes one average backscatter value (target strength) to obtain a flow rate value. It does not integrate a sector of the flare or normalizes the value by the volume or area. At deeper sites the footprint is larger and thus possibly able to cover more single bubble streams at once. This could be a reason of having stronger flares and subsequently higher flow rate values.

R3.11 Line 482: why should there be more bubbles only because the flare does not have a single source?

Reply: This part was removed from the revised manuscript.

R3.12 Line 494: the gas hydrate dissociation would not necessarily follow the 500m isobaths but be related to ocean currents and their changes. They determine the temperature distribution. The area is known to encounter upwelling. Still I agree, it would not be locally confined; even so currents in canyons and at the outflow of the JdF St could be specific.

Reply: The discussion on gas hydrate dissociation and the 500 m isobath was simplified and downscaled overall, as it is not the focus of our contribution.

R3.13 Line 503 ff: there are publication for the area, e.g. from the Coal Oil Point (Mau et al, 2007) or onshore Hydrate Ridge (Rehder et al., 2002). Overall the short discussion seems rather superficial and has little overlap with the dataset. You might want to consider taking this out.

Reply: As suggested, this part was removed.

R3.14 Line 508 ff: This list partly seems to be a repetition of things just said. Number 5 is what fits best to your data set. Number 1 is tackled in Römer et al., (2016) and other literature; Numbers 2 – 4, I would attach to the data evaluation while for #5 the new data set could be very valuable. Especially on the shelf and upper slope seasonal changes might play a significant role (upwelling; undercurrent; river discharge...)

Reply: We modified the section as suggested to avoid too much overlap/duplication.

R3.15 Abstract: You might want to add a range/error to the methane flux data.

Reply: Done as suggested.

R3.16 Introduction: The first paragraph is incoherent (e.g. active, passive, Arctic margin is a mix) and lists all “hot” topics vaguely related to methane fluxes. Less might be more and would allow being more precise. E.g., the CH₄ outflow has a rather very limited impact on the carbon in marine sediments, “Natural flux of gases” (37) is a very vague expression.

Reply: We made a few modifications, but kept at least a spread of references to other regions to show the abundant previous work and importance of assessing methane fluxes.

R3.17 Line 78: natural flux of methane: out of the sediment or quantifications within the sediment

Reply: We included “across the sediment/water interface” to address this issue.

R3.18 Line 89-94: The text refers to data are on pore water measurements of CH₄ fluxes (dissolved and /or within the sediment?). This might be redundant to this data set.

Reply: This section points towards one of the very few long-term studies and we feel it is thus warranted to keep the citation and topic included in the introduction. We modified the text to indicate why we feel that those few longer-term observations provide meaningful constraints.

R3.19 Literature: It might be worthwhile to mention Coal Oil Point, which is a well investigated vent site on the shelf (California) in terms of methane transport and fluxes (e.g. Clark et al., 2000; Mau et al., 2007). Further, there is a publication by Rehder et al., 2002 regarding ocean-atmosphere gas fluxes of methane at Hydrate Ridge and the shelf area. McGinnis et al. (2006) – Black Sea; Westbrook et al. (2009) Svalbard (e.g.)

Reply: This portion of the manuscript was removed in the revision.

R3.20 Methods: Without any information on the methods I find it very hard to read or assess the results and Fig. 3.

Reply: We modified the figure caption [and added subtitles to the graphs] and we hope this makes it easier to read the figure.

R3.21 Appendix Line: 571: The temperatures are higher : < 500 m: 5-9°C; 500 – 1000 m: 3.5 – 5°C

Reply: We added a new first-order function as approximation to define temperature as function of water depth. This equation does not fully address seasonal or longer-term changes, but provides a more realistic temperature at the flare sites.

R3.22 Fig. 1 There will be distinctions between the shelf, 250 -1000 m, and >1000 m. Apart from the 500 m isobar these depth ranges are not resolved in the Figure. In the right panel one could leave out the vent sites to better visualize the tracks and change the gray lines to a slightly darker gray.

Reply: Figure 1 was updated to address this comment.

R3.23 Fig. 3 The figure caption is rather short. It might just be me but I had trouble understanding Fig. 3 at this point and thus had difficulties following the normalization. Percentages are really difficult to follow up.

Reply: We modified the caption, hoping it will improve the overall understanding.

R3.24 Line 203-204: with the very low number of lines crossing at the DF - apart from the area offshore Vancouver Island - I would restrain from this statement

Reply: Overall, yes, few lines exist, but the northern part had several relevant lines with no venting, and the southern part has a couple of lines with some venting, and so we feel comfortable to keep this statement, though we added the word “possibly.”

R3.25 Acoustic quantification of gas emissions: Bubbles below 500 m might have a gas hydrate coating (e.g. Rehder et al., Sauter et al.). The coating should alter the response to soundwaves compared to bubbles without. Is this difference included? Does your model include the different bubbles size distribution encountered at vent sites in different depth ranges?

Reply: We now show the actual calculations for clean and dirty bubbles in the revised manuscript. It provides some measure for uncertainty. Overall, we use the “clean bubble theory” in all reported values. We now show that using dirty bubbles shifts the flow rate estimate less than 20%.

R3.26 Line 282ff: significant variations in the flow rate estimate.. may represent real changes.. but could also be a result due to noise in the data. This is a statement that completely undermines your calculations. What’s the error introduced by the quality of the data?

Reply: We rephrased the sentence in question to better reflect the confidence we have in our results. The data quality among these many cruises and years of acquisition may indeed vary, but unfortunately data quality cannot be quantified. We were very strict when assessing the flare-data and only use data that is free of obvious noise (fish or plankton layers).

R3.27 Fig. 5 The comparison between the flow rates and the tidal cycle is not very clear – you might want to add another cycle. (Römer et al., 2016)

Reply: We introduce a whole new tidal-model. Please see new method section.

R3.28 How about the currents which are much stronger at depth < 500 m? The dissolution should be faster as should be the bubble dispersion. This is taken into account in your calculations, is it not?

Reply: Bottom currents are not considered.

R3.29 Margin-wide gas flux: Line 352 This assumption might hold some difficulties: the clustering along the canyons might not fit the overall “physical constraint of the geographical region”. The idea of rise rates being the same all over the shelf is a daring thesis as, e.g., the current regimes and river discharges vary considerably in this geographical region.

Reply: As mentioned previously, we restrain from too much interpretation of the regional distribution of flares and links to canyons or other geographical features. However, rise-rates may indeed vary when currents or other topographic features affect the venting. As said by the reviewer: a good topic for a thesis – but outside the scope and aim of our work.

R3.30 Line 357: It seems like 1600 m should be your maximum depth since the data set is too scarce below this depth.

Reply: As seen in the revised Figure 7, we only show data up to 1600m. Since this was not clear, we added this also to the text.

R3.31 Line 359: Is the tidal forcing included in your flux calculations?

Reply: Tidal forcing is now introduced. See new Method section.

R3.32 Lines 369 ff – what defines minimum and maximum flow rates where before there was not such a thing? For the first time there is a 2 order of magnitude variation for your rates.

Reply: We have updated discussion on errors, statistical spread and overall uncertainty.

Reviewers' comments:

Reviewer #1 (Remarks to the Author):

Each of the points raised in the initial review have been satisfactorily addressed by the authors. I expect the manuscript will be well received and highly cited by the research community. I have no reservations regarding the publication of the manuscript.

Reviewer #2 (Remarks to the Author):

In my initial review of this work, I felt that there were two substantial concerns that required attention prior to a subsequent review. Of the two, the authors have satisfactorily answered one of them. The other (the original first concern, described below) and several other comments should be addressed. I believe this will require a second major revision.

The first of these concerns was the lack of any knowledge of a bubble size distribution. On this issue, the authors now apply a bubble size distribution from ROV video observations. Their description of how they extract the bubble size distribution appears to be limited to this sentence "A new bubble size distribution was defined for a gas flare (Figure 9a) seen close to the bubble sonar location" (lines 674-675) and Figure 9a. To me, this description is inadequate and leaves to many questions: was this stereo imagery? Were the cameras calibrated in some way? What was the uncertainty in range between bubble and camera, and how does this propagate to the uncertainty in the bubble radius? The authors also state that flow rate estimation can "vary by up to a factor of 3 when applying the different distributions on the same acoustic data set". I don't doubt this, and it can probably be higher when the uncertainty is greater. This is not to argue that these kind of approaches can't be done – this wouldn't be the first time, after all - but simply to argue that the authors have not adequately described how they arrive at their bubble size distribution or how they arrive at their 'factor of 3'. They DO appear to have gone back and revised their work, but I don't think they've adequately reported it.

My second major concern was related to the emphasis the authors have placed on the 500-m isobaths, and whether this was an observational artifact rather than a real effect. They have now reduced (although not eliminated) their attention to this, although they still don't discuss other similar 'spikes' in their histograms at the 200 m and 950 m isobaths. I understand the intrigue of the dissociating methane hydrates, but am disappointed that they don't attempt to equally address these other similar anomalies.

Other comments:

This work seems to be a more comprehensive version of the Johnson paper (reference 38) who studied the same area but didn't go as deep into the archive, analyzing only 139 flares? It is good that the paper is referenced, but it would be helpful to readers to more readily know that this analysis from a similar region has also been conducted.

The authors state on lines 197-200 that "Simple extrapolation of the observed venting into uncharted areas is considered misleading because locations are likely not randomly distributed; they may rather follow geological trends or may be linked to zones of high biological productivity and sedimentation rates providing organic carbon for microbial activity to produce methane gas". I agree. However, the authors seem to be neglecting this statement by doing this extrapolation, stating the results of the extrapolation in the abstract and lines 404-414. Is the abstract misleading, then? This inconsistency should be resolved.

On lines 275-277, the authors state that they use a 'clean bubble' model, which basically means that the gas at the bubble boundary can slip along with the seawater moving past it. But the area

in question is more than 1000 m deep, and so presumably the bubbles have a hydrate coating? The bubbles should be considered dirty, and these models should be re-run. On that note, an alternative explanation for the slow rise velocities (which presumably aren't measured at the seabed) is that the bubble size distribution is in error.

On lines 278-280, the authors state that they integrate the flow rates over 12 hours to account for the tidal cycle. But aren't these mixed semidiurnal tides (as can be seen at this same site in Romer et al., the author's reference 39)? The integration should be done over a 24 hour period rather than a 12 hour period.

Line 398: Methane density: what EOS did the author's use (methane is not an ideal gas). More information needed.

Regarding statistics: On line 331-332 the authors state: "Since the average deviations for the integrated flow rates over one tidal cycle is larger than the average value itself, the lower bound of any flow estimate is mathematically negative." This is simply not true. A person could arrive at this (erroneous) conclusion by assuming that the flow rates are normally distributed, but of course that can't be the case. There are many distributions which include only positive values (exponential, log-normal, Rayleigh, Weibull, etc.). This should be changed in the text, but it also means that each and every +/- value should be changed. A more thorough statistical analysis is required in order to accurately/appropriately assess the upper and lower bounds of these data.

The author's estimates for the range of flux rates on lines 588-589 only account for different normalizations (which, see earlier, they have said are misleading). The range only accounts for the different flux-rates, but not for any of the observations. So the resulting range should probably be much larger? Further, there is a question the authors have brought up, which should increase the uncertainty even further, this "critical question is whether the process controlling flare occurrence is a truly random process (which is the assumption in all these normalizations) or controlled by other factors as discussed below and if such normalizations are thus legitimate" (see lines 242-244). This needs to be addressed in the manuscript more quantitatively for the resulting methane flux estimates to be useful.

One lines 681-682, the authors state that "The rise rates defined from the acoustic data (Figure 9b) are all close to each other (varying from 13 cm/s 682 to 23 cm/s), which point to similar bubble-size distributions between these five vent regions." That's not actually true – above 2 mm dia, the rise rates for bubbles do not vary greatly and are mostly in the vicinity of 20 cm/s – so the author's can probably say that the acoustically observed bubbles are probably mostly above this size, but this doesn't say much about the actual distribution (e.g., how many 5 mm bubbles there are, or how many 10 mm bubbles, etc.)

Reviewer #3 (Remarks to the Author):

Review M Riedel

the authors put considerable effort into improvising the calculations and discussion on the uncertainties related to their model and flux calculation, which certainly helps the credibility of the paper. They discarded most of the aspects their data or current interpretation of the data would not support sufficiently, which also improved the manuscript.

By adding a comprehensive discussion on the data reliability, which is clearly necessary, the main aspect of the manuscript shifted further towards method development and data handling of EK60 and multibeam data for gas flux calculations. The interpretation of the impressive compilation of data on vent distribution along the Oregon margin is still very limited despite the abstract

promising differently as do lines 97-98. The text on the distribution of vent sites is rather general and mostly points towards the limits of the data set (Line 184-217). If the data interpretation is the major point and interest to the reader, the paper has improved.

The addition of a bubble size distribution model, the quantification of "clean" and "dirty" bubbles, and the addition of a tidal model improve the instantaneous flow rates and those integrated over time for a single flare to some extent.

The flux rates and budgets are what I am still skeptical about. Based on the nature of the data I am still very skeptical on the error ranges mentioned with respect to the individual flare flow rates: Checking Figure 9 (Supplement): there is a large number of bubbles between 0.1 and 0.3 cm (a). In (c) there is no difference in rise speed; however, there are two orders of magnitude difference in volume between these bubbles (0.004 cm³ compared to 0.113 cm³). How does this effect the calculation? On top of this, flares with high bubble density in a column have a different rising behavior compared to single bubbles – unfortunately I do not know the significance of this effect. Figure 9b lacks any rise rates above 400 m. On the shelf relative pressure changes, flow rates and possibly venting mechanisms as well as current speeds are very different. One should assume the bubble sizes distributions being rather different

Line 672: Flow rate estimations vary by a factor of 3? Wouldn't this lead to more than 200%

The calculation of flow rates on the single flares and their extrapolated budget likely have large error bars, especially for the shelf area where there is no bubble distribution known at all, nor rise rates. There will be a large error propagation if applying yet a second normalization (bathymetric depth) were a random rather than a tectonically determined distribution is assumed and commented on by the authors: Line 242-244: .. the critical question is whether the process controlling flare occurrence is a truly random process (which is the assumption in all the normalizations) or controlled by other factors as discussed below and if such normalizations are thus legitimate. With the ship track and bathymetric depth normalization being similar, the sampling bias might be less of an issue. However, your map clearly shows that the vents/flares are not normally distributed and the mapped areas cover less than 10% of the shelf and an average of 20% between 250-1000 m (footprint). Line 531 – 533 point towards the same issue: "2/3 of these gas vents around 500 m water depth are occurring between 47.5° and 48.25° latitude"

Comment on R 3.6d: The many sites at Clayoquot significantly increase the number of flares you find in this depth range which will be part of the depth normalization. Even excluding repetitions this might not be representative for this depth range along the total margin, as there might just not be another Clayoquot in the areas not having been sampled.

A few other things:

I would still suggest referring to the data set from Römer et al. 2016 for the tidal modulation. In Figure 5 b the highest values is not necessarily related to low tide. It's a good figure to show but the argument is Römer's or others before them, I would think. Line 293.. ."set was acquired "and confirmed a strong tidal influence on the flow intensity. An example..

Line 48 – 50 - later on the authors argue that the distribution might also be nutrient related. Here is tectonically only – canyons do cut through quite a number of layers...

Table 4 does not mention that the rates are in situ rates. This is very easy to confuse as it is rather uncommon. Binning in situ fluxes seems difficult

Line 226 – starting at 10 m water depth. In the reply to the reviewers the author states that the

shallowest point of measurement is 40 m. Why do you start at 10 m?

Line 288 – The statement "the variability of repeated measurements at a single site might be due to noise etc." might not increase trust in a budget that is inferred from these data. Line 293 indicates that your differences in flow are not only related to noise! Did you apply your tide model to the flares you visited repeatedly during different years and seasons? Does this match to some extent?

Reply to individual review comments

Distributed natural gas venting offshore along the Cascadia Margin

M. Riedel, M. Scherwath, M. Römer, M. Veloso, M. Heesemann, and G.D. Spence

We would first like to thank the reviewers for their commitment for a second review and all additional comments provided which were useful and guided us through improving the manuscript work. Below, we outline our replies to the reviewer comments and explain all changes made to the manuscript in response to the comments raised. As a number of requirements limit the amount of material to be presented within the main article (5,000 words, 10 display items max including Figures and Tables), we eliminated one of the original figures (old Figure 5) and switched the call-out of tables so that 2 Tables (#4 & #5) and 2 Figures (#8, #9) are only occurring as part of the Methods section.

Reviewer #2 (Remarks to the Author):

In my initial review of this work, I felt that there were two substantial concerns that required attention prior to a subsequent review. Of the two, the authors have satisfactorily answered one of them. The other (the original first concern, described below) and several other comments should be addressed. I believe this will require a second major revision. The first of these concerns was the lack of any knowledge of a bubble size distribution. On this issue, the authors now apply a bubble size distribution from ROV video observations. Their description of how they extract the bubble size distribution appears to be limited to this sentence “A new bubble size distribution was defined for a gas flare (Figure 9a) seen close to the bubble sonar location” (lines 674-675) and Figure 9a. To me, this description is inadequate and leaves to many questions: was this stereo imagery? Were the cameras calibrated in some way? What was the uncertainty in range between bubble and camera, and how does this propagate to the uncertainty in the bubble radius?

REPLY: We significantly augment our description on how we derive the bubble size distribution. The video was made by the ROV ROPOS. An inverted funnel (basically a kitchen tool) with a scale was directly held into a bubble stream at Clayoquot slope so that bubbles are seen entering the funnel close to the scale. The distance between funnel and camera is

determined accurately to within 10cm, as the robotic arm of the ROV is used to manipulate the funnel and its dimensions are well known to define the distance. Additionally, the standard technique of calibrated double laser spots is used at the beginning of the deployment, so the lateral distances of the image is known. The cameras are routinely color-calibrated by ROPOS during dives while in the water, and a general pre-dive calibration on deck is also performed. The camera of ROPOS sits in such a position that the view to the target (i.e. the funnel) is at a 45° angle. Imagery was rectified to overcome image distortion prior to measuring bubble sizes and rise-rates. These measurements are in situ, at the seafloor and thus applicable to the task at hand. They were not measured at greater height above seafloor.

Many of the detailed questions raised by the reviewer about error-propagation cannot really be answered as there is no option for verification of any of the measurements by a different technique (or 2nd camera). However, the size of a bubble may be determined with a relative uncertainty mostly linked to image resolution (and quality, i.e. blur). However, this uncertainty (let's assume for the sake of this argument an error of 0.5 mm) is equally applied to all bubble sizes. Therefore, this uncertainty does not alter the shape of the distribution (e.g. number of bubbles at any given diameter) and therefore it does not affect the estimated flow rate.

The author's also state that flow rate estimation can “vary by up to a factor of 3 when applying the different distributions on the same acoustic data set”. I don't doubt this, and it can probably be higher when the uncertainty is greater. This is not to argue that these kind of approaches can't be done – this wouldn't be the first time, after all - but simply to argue that the authors have not adequately described how they arrive at their bubble size distribution or how they arrive at their ‘factor of 3’. They DO appear to have gone back and revised their work, but I don't think they've adequately reported it.

REPLY: A factor of three difference in flow rate for a particular flare was determined empirically. We were using the methods by Leifer et al. (2000), Leifer&Patro (2002), and Mendelson (1967) and for a given bubble size distribution, the estimated flow rates from these three theories are linearly dependent on each other as described in the manuscript. If the bubble size distribution is modified (e.g. using the distribution published in Veloso et al. (2015) instead of the distribution newly determined at Cascadia, then all these estimates go up by a factor of

three. A different factor is probably found if yet another bubble-size distribution (from elsewhere in the world) is applied in the calculations. This is the fundamental uncertainty in the flow rate calculation and we have no mathematical way to determine a “correct” uncertainty that reflects the basic lack of sufficient observations and comparable studies at Cascadia. We therefore only refer to this one scalar (factor of three) as the bubble size distribution by Veloso et al. (2015) is available for the reader including the tool to re-calculate the flow estimates, and thus the values reported could in theory be reproduced by any interested reader (especially since all acoustic data are also publically available).

In order to account for this general problem in the flow rate estimation and to address (both of) the reviewer’s comment, we clarified the text and introduced a new section in the Methods on how we derive our total uncertainty. After careful re-considerations of the (many) factors of uncertainty in the method employed, we have newly estimated our total uncertainty. While originally we reported only the largest single uncertainty (factor of three from bubble size distribution error), we now describe the total uncertainty in the method by a multiplication of the individual effect introduced from all (reasonable) parameter variations. Thus, any given flow rate for any flare listed in Table 3 can be larger by a factor of 4.8, or lower by a factor of 0.18. All details to the origin of the numerical values are given in the revised manuscript.

My second major concern was related to the emphasis the authors have placed on the 500-m isobaths, and whether this was an observational artifact rather than a real effect. They have now reduced (although not eliminated) their attention to this, although they still don’t discuss other similar ‘spikes’ in their histograms at the 200 m and 950 m isobaths. I understand the intrigue of the dissociating methane hydrates, but am disappointed that they don’t attempt to equally address these other similar anomalies.

REPLY: The 500 m isobath and possible link to gas hydrate dynamics was downplayed but not eliminated as there is evidence for possible warming induced gas hydrate dissociation and links to increased gas fluxes, as described in the literature. The primary reason to keep this 500m isobath in this revised manuscript, however, is that it divides the two domains of ‘clean’ and ‘dirty’ bubble theory; this is a new addition to the paper based on the reviewer’s comment below.

In terms of additional level of interpretation to secondary spikes in the data set, we have added some text in the revised manuscript to describe a possible origin of the spikes. While the spike at ~950 m water depth is associated to only three clusters of flares at the two large canyon systems and thus being rejected as reflecting a regional process, the spike at around 200 m (actually the spike is located at 160 m) cannot be as easily dismissed. When plotting all flares in this depth range, they only spread between 46.8°N and 48.9°N but they do not occur along a single isobath parallel to the shelf edge (as e.g. the flares at 500 m water depth), but also occur linked to incised channels on the broad shelf around the Juan de Fuca Strait entrance. Without detailed interpretation of high-resolution sub bottom profiler data across these flare sites, the nature of the spike in flare occurrence cannot be verified. We could speculate on the origin (outcropping strata, limit of ice-extent across the shelf, thus an erosional unconformity) but these are non-substantiated and would require too much additional detail that it would distract from the main purpose of our contribution.

Other comments:

This work seems to be a more comprehensive version of the Johnson paper (reference 38) who studied the same area but didn't go as deep into the archive, analyzing only 139 flares? It is good that the paper is referenced, but it would be helpful to readers to more readily know that this analysis from a similar region has also been conducted.

REPLY: While the paper by Johnson et al. (reference #38, cited 10 times throughout our manuscript) is an important contribution (including others from his research group, which also are included in the references), we restrain from further highlighting one specific contribution. The work by Johnson et al was aimed more on regional occurrence than on flow rates and thus reports on different findings.

The authors state on lines 197-200 that “Simple extrapolation of the observed venting into uncharted areas is considered misleading because locations are likely not randomly distributed; they may rather follow geological trends or may be linked to zones of high biological productivity and sedimentation rates providing organic carbon for microbial activity to produce methane gas”. I agree. However, the author's seem to be neglecting this statement by doing this

extrapolation, stating the results of the extrapolation in the abstract and lines 404-414. Is the abstract misleading, then? This inconsistency should be resolved.

REPLY: The reviewer is correct in that there appears to be an apparent contradiction in our line of arguments. This was also pointed out by reviewer #2. We basically report three different values for the margin-wide flux: (1) based on the ~400 estimates of ~300 individual flares projected onto all identified flares, (2) depth-based normalizations (as proposed and applied by Johnston et al (Reference #38) and (3) normalization by foot print (new method). Applying assumption (1) would be honoring the observation of non-normal distribution of vents, i.e. “what we see is all there is”. This is representative of a total lower limit of gas flux (which we believe is still a useful and NEW contribution). Applying any of the two normalizations would force the distribution of vents being normally (randomly) distributed, thus giving a second extreme (and likely maximum) view on the total methane flux. If one is to believe in a normal (random) flare distribution, then normalization by footprint would be a better representation as it more truly honors the data sources and their ability to map gas flares.

The “true” answer for the total margin-wide gas flux is probably in between these extremes (where each value is still uncertain by the nature of treating the physics of the problem as described (and prescribed) in the newly revised manuscript). We do offer these different flow rate values, and propose the footprint normalization (and also tidal integration) as a more representative normalization technique.

We also chose to report the values from depth normalization in the abstract, as previous literature from Cascadia also used such normalization scheme. With this, we hope that the inconsistency in text and values is removed.

One lines 275-277, the author’s state that they use a ‘clean bubble’ model, which basically means that the gas at the bubble boundary can slip along with the seawater moving past it. But the area in question is more than 1000 m deep, and so presumably the bubbles have a hydrate coating? The bubbles should be considered dirty, and these models should be re-run. On that note, an alternative explanation for the slow rise velocities (which presumably aren’t measured at the seabed) is that the bubble size distribution is in error.

REPLY: We thank the reviewer for highlighting this important difference in bubble occurrence

mode. In the original manuscript we have stated the impact on using either 'clean' or 'dirty' bubbles (original line 688), but did not further engage in calculations. We have therefore introduced this difference more rigorously and applied a cutoff at the 500 m isobath, which is used as proxy for the upper limit of gas hydrate stability along the Cascadia margin. That means, flares in water depths > 500 m are treated with the 'dirty' model (i.e. a reduction to 82% of the original instantaneous flow rate calculated) and flares in shallower water (< 500m) remain unchanged to the previous calculations. Thus, our regional extrapolated values have changed. In the revised manuscript, we included this change in approach accordingly.

On lines 278-280, the authors state that they integrate the flow rates over 12 hours to account for the tidal cycle. But aren't these mixed semidiurnal tides (as can be seen at this same site in Romer et al., the author's reference 39)? The integration should be done over a 24 hour period rather than a 12 hour period.

REPLY: The work by Römer et al. (2016), Figures 6 and 7, showed a strong diurnal tidal influence but the modes of fluctuations as gas flare activity are symmetrical over 12 hours. The flare activity always falls back to the same minimum after each (different) maximum, therefore the unsymmetrical shape of the tide does not influence the symmetry of the gas emission activity over the period of 24 hours. That means, integration over 12 hours with subsequent multiplication by a factor of two to account for a complete day results in the same total flow rate as if integration of 24 hours is done. We therefore do not change the reported method and values.

Figure 6 by Römer et al. (2016).

Line 398: Methane density: what EOS did the author’s use (methane is not an ideal gas). More information needed.

REPLY: We added new text to clarify this statement in the revised manuscript. Density of methane gas as a function of pressure (and temperature) is calculated with the MATLAB® codes derived and published by colleagues at GEOMAR in form of the “SUGAR toolbox” (Kossel et al., 2008). Although the reference was given in Table 3, but it was obviously too strongly hidden from obvious recognition.

Regarding statistics: On line 331-332 the authors state: “Since the average deviations for the integrated flow rates over one tidal cycle is larger than the average value itself, the lower bound of any flow estimate is mathematically negative.” This is simply not true. A person could arrive at this (erroneous) conclusion by assuming that the flow rates are normally distributed, but of course that can’t be the case. There are many distributions which include only positive values (exponential, log-normal, Rayleigh, Weibull, etc.). This should be changed in the text, but it also means that each and every +/- value should be changed. A more thorough statistical analysis is

required in order to accurately/appropriately assess the upper and lower bounds of these data.

REPLY: We thank the reviewer again for this helpful comment. After going back through the data and evaluating the statistics, we agree that probability distributions other than normal should be applied. We have chosen the log-normal distribution to derive new mean values and standard deviations for the shallow-, medium-, and deep-water settings. Following the work by Limpert et al. (2001) we transform the data (flow rates) into the Log-domain, derive statistics for a normal distribution, and back-transform the mean and standard deviations. The new mean is thus the originally reported median values of flow rates and new minimum and maximum values are defined. Thus, all min/max ranges were updated, the median values being used for extrapolation and summation of flow-rates across the margin.

Limpert, E., Stahel, W.A., abbt, M., 2001. Log-normal distributions across the Sciences: keys and clues, BioScience, 51(5), 341 – 352.

The author's estimates for the range of flux rates on lines 588-589 only account for different normalizations (which, see earlier, they have said are misleading). The range only accounts for the different flux-rates, but not for any of the observations. So the resulting range should probably be much larger?

REPLY: We have modified our approach of estimating the range in flow rate (and margin flux) in the revised manuscript. The different types of normalizations are compared with no preference given to any number at this point.

Further, there is a question the authors have brought up, which should increase the uncertainty even further, this “critical question is whether the process controlling flare occurrence is a truly random process (which is the assumption in all these normalizations) or controlled by other factors as discussed below and if such normalizations are thus legitimate” (see lines 242-244). This needs to be addressed in the manuscript more quantitatively for the resulting methane flux estimates to be useful.

REPLY: In order to address this issue, we modified the text and generally define our approach as to be based on random flare distribution assumptions. In the discussion, we briefly outline how one could estimate the “true” number of vents present (or in other words, what was missed)

by e.g. using Horvitz-Thompson estimators. Yet, this task is way beyond the scope of our study presented here. We would love to continue research into this question and will collaborate with other scientists familiar with these advanced statistical methods and hopefully will in the future be able to define a H-T estimator (or equivalent) approach to gas flare detection, which adopts the many unique issues around gas flare detection (including e.g. tidal modulation and missing a flare due to “unlucky” crossings of flare sites).

One lines 681-682, the authors state that “The rise rates defined from the acoustic data (Figure 9b) are all close to each other (varying from 13 cm/s to 23 cm/s), which point to similar bubble-size distributions between these five vent regions.” That’s not actually true – above 2 mm dia, the rise rates for bubbles do not vary greatly and are mostly in the vicinity of 20 cm/s – so the author’s can probably say that the acoustically observed bubbles are probably mostly above this size, but this doesn’t say much about the actual distribution (e.g., how many 5 mm bubbles there are, or how many 10 mm bubbles, etc.)

REPLY: We thank the reviewer for this additional insight and correction of our statement. We have changed the manuscript accordingly. Thus, the rise rates from the different flare locations show similar rise rates, which means that the bubble sizes are most likely larger than the lower cut-off. This does not contradict our basic conclusion, in that the bubble size distribution (especially the lower limit) we newly defined is representative for the region.

Reviewer #3 (Remarks to the Author):

The authors put considerable effort into improvising the calculations and discussion on the uncertainties related to their model and flux calculation, which certainly helps the credibility of the paper. They discarded most of the aspects their data or current interpretation of the data would not support sufficiently, which also improved the manuscript. By adding a comprehensive discussion on the data reliability, which is clearly necessary, the main aspect of the manuscript shifted further towards method development and data handling of EK60 and multibeam data for gas flux calculations. The interpretation of the impressive compilation of data on vent distribution along the Oregon margin is still very limited despite the abstract promising differently as do lines 97-98. The text on the distribution of vent sites is rather general and mostly points towards the limits of the data set (Line 184-217). If the data interpretation is the major point and interest to the reader, the paper has improved. The addition of a bubble size distribution model, the quantification of “clean” and “dirty” bubbles, and the addition of a tidal model improve the instantaneous flow rates and those integrated over time for a single flare to some extent.

REPLY: We thank the reviewer for these general positive remarks on our revised manuscript.

The flux rates and budgets are what I am still skeptical about. Based on the nature of the data I am still very skeptical on the error ranges mentioned with respect to the individual flare flow rates: Checking Figure 9 (Supplement): there is a large number of bubbles between 0.1 and 0.3 cm (a). In (c) there is no difference in rise speed; however, there are two orders of magnitude difference in volume between these bubbles (0.004 cm³ compared to 0.113 cm³). How does this effect the calculation?

REPLY: The entire approach to estimating flow rate uncertainty, especially from errors in the bubble size distribution, has been modified and a new section is included in the further revised manuscript.

On top of this, flares with high bubble density in a column have a different rising behavior compared to single bubbles – unfortunately I do not know the significance of this effect. Figure 9b lacks any rise rates above 400 m.

REPLY: Yes, this is true – unfortunately we have not investigated (and do not know of any such study available) on vents in shallower water.

On the shelf relative pressure changes, flow rates and possibly venting mechanisms as well as current speeds are very different. One should assume the bubble sizes distributions being rather different

REPLY: Yes, the reviewer is correct in that the shelf environments are rather different and that one could expect changes to bubble size behaviors (and how such bubbles are taken up by currents, etc.). However, we do lack such observations necessary to advance the data base of bubble-size distributions in Cascadia. Instead we define new error estimates from applying a uniform distribution (see new section in Methods).

Line 672: Flow rate estimations vary by a factor of 3? Wouldn't this lead to more than 200%

REPLY: We obviously did not change this value correctly in the last revision round of revision, to our own frustration. In this newly revised manuscript, uncertainty values and ranges in estimated values are newly defined. See our replies to Reviewer #2

The calculation of flow rates on the single flares and their extrapolated budget likely have large error bars, especially for the shelf area where there is no bubble distribution known at all, nor rise rates. There will be a large error propagation if applying yet a second normalization (bathymetric depth) were a random rather than a tectonically determined distribution is assumed and commented on by the authors: Line 242-244: the critical question is whether the process controlling flare occurrence is a truly random process (which is the assumption in all the normalizations) or controlled by other factors as discussed below and if such normalizations are thus legitimate. With the ship track and bathymetric depth normalization being similar, the sampling bias might be less of an issue. However, your map clearly shows that the vents/flares are not normally distributed and the mapped areas cover less than 10% of the shelf and an average of 20% between 250-1000 m (footprint).

REPLY: We have augmented the Method section by including proper uncertainty estimation (see details in above response of Reviewer #2) to account for (both) the reviewer(s) concerns. We also discuss in more detail the validity of using our normalization technique and possible deviations from the assumed “random” distribution.

Line 531 – 533 point towards the same issue: “2/3 of these gas vents around 500 m water depth are occurring between 47.5° and 48.25° latitude”.

REPLY: The reviewer is correct in all statements that errors are being propagated from individual local flow estimates (described and defined more thoroughly now) to margin-wide distributions by normalizations. We basically report three different values for the margin-wide flux: (1) based on the estimates of ~400 individual flow rates projected onto all found flares sites (~1030) binned by water depth cut-offs, (2) depth-based normalizations (as proposed and applied by Johnston et al (Reference #38) and (3) normalization by foot print (new method). Applying assumption (1) would be honoring the observation of non-normal distribution of vents, i.e. “what we see is all there is”. This clearly can only be representative of a lower limit of total gas flux (which is still a very useful and NEW contribution). Applying any of the two normalizations would force the distribution of vents being normally (randomly) distributed, thus giving a second extreme (maximum?) view on the total methane flux. If one is to believe in a normal (random) flare distribution, then normalization by footprint would be a better representation as it better honors the data sources and their ability to map gas flares.

The “true” answer (if such exist) for the total margin-wide gas flux is probably in between these extremes (where each value is still uncertain by the nature of treating the physics of the problem as described (and prescribed) in the newly revised manuscript). We do offer these different values and proposed footprint normalization with tidal integration as more representative normalization technique.

Comment on R 3.6d: The many sites at Clayoquot significantly increase the number of flares you find in this depth range which will be part of the depth normalization. Even excluding repetitions this might not be representative for this depth range along the total margin, as there might just not be another Clayoquot in the areas not having been sampled.

REPLY: Clayoquot, the site of the ONC node, is measured in our regional statistic only once. Repeated flare-estimates (averaged over numerous repeat visits) helped to ascribe a more robust flow estimate for this vent (and thus water depth range). And the reviewer may well be right in that, along Cascadia, there may only be this one vent site at Clayoquot and we were lucky to find it.

A few other things:

I would still suggest referring to the data set from Römer et al. 2016 for the tidal modulation. In Figure 5 b the highest values is not necessarily related to low tide. It's a good figure to show but the argument is Römer's or others before them, I would think. Line 293..."set was acquired "and confirmed a strong tidal influence on the flow intensity.

REPLY: The reviewer is correct in that we only confirm previous findings and thus modified the text accordingly. We also felt that the original Figure 5 was not adding much to the overall discussion and due to the need to reduce overall number of display items, this figure was deleted.

Line 48 – 50 - later on the authors argue that the distribution might also be nutrient related. Here is tectonically only – canyons do cut through quite a number of layers...

REPLY: We augmented the introduction (and discussion) around the possible controlling factors of vent distributions.

Table 4 does not mention that the rates are in situ rates. This is very easy to confuse as it is rather uncommon. Binning in situ fluxes seems difficult

REPLY: We modified the caption accordingly.

Line 226 – starting at 10 m water depth. In the reply to the reviewers the author states that the shallowest point of measurement is 40 m. Why do you start at 10 m?

REPLY: Unfortunately, to our own frustration, this was a left-over from the original manuscript and was now fully removed in the 2nd revision.

Line 288 – The statement "the variability of repeated measurements at a single site might be due to noise etc." might not increase trust in a budget that is inferred from these data. Line 293 indicates that your differences in flow are not only related to noise! Did you apply your tide model to the flares you visited repeatedly during different years and seasons? Does this match to some extent?

REPLY: We have modified this portion of the text to better demonstrate measurement uncertainty, choices of flare-data, exclusion of portions that are noisy, etc., and thus strengthen the trust into the method employed.

Reviewers' comments:

Reviewer #2 (Remarks to the Author):

I think the authors have done a good job addressing the concerns raised in the two earlier reviews, and have sufficiently modified the manuscript to make it acceptable.

Manuscript Riedel et al.,

General comments:

The shortening of the text, the error information and the introduction of a chapter on uncertainty did considerably improve the manuscript.

I would still like to point out a couple of general comments; two of them have been mentioned earlier but I still find them difficult and one of them is new – and I am sorry for only bringing this up now.

+ Random versus geographically controlled distribution: the “Results” mirrors this issue. While one paragraph discusses the clustering of venting the next paragraph basically states that this is interesting to know but that a random pattern is applied. Even if it does not erase the problem you might want to separate the two topics. It makes it hard to believe in the calculations and it will not help discussing on the causes of the vent distribution.

+ Line 199 - 202: "... the shelf regions become more amplified (Fig. 4c)." Is this also true if you ignore the large amount of vents in the JdF strait area?? Is it this particular environment that is amplified or is it the overall shelf. Might be worth a try? How does this relate to the 500 m isobar?

+ Line 394 - 399 I cannot quite make out the argument here. As it seems to me, the authors mention a method for random distribution, do not use it and conclude that there is a random distribution. As it is this argument /paragraph does not seem convincing.

+ Line 415 – 423 "reveals that 2/3 of these flares are occurring between 47.5° and 48.25° latitude." The authors have already commented on this.

+ The bubble size issue: Is there no data set (MBARI?) for any shallower vent site. E.g. Lines 335 – 336, Line 432 - 433. Regarding the large difference in pressure and change in relative pressure between the deep sites and the shallow sites, it is very unlikely that bubble size maxima are similar in both environments. Referring to Fig. 11 in Veloso et al., 2015, the bubble size maxima at Clayoquot of 1 – 3 mm are low compared to any of the other sites shown in this figure. The Santa Barbara Channel site might be rather shallow and maxima are slightly higher; the Svalboard data show 4 -6 mm maxima, however, there are thermogenic sources as well. Why choose this data set and use it for all depth not for the shelf only? Anything above 4 mm significantly changes the bubble rise speed (your Fig. 8c).

+Would it be worthwhile to calculate the shelf data using one of the shallower data sets while keeping your data for the deep vents?

+In Vescos paper there are some other bubble size distributions all of which have higher average bubble sizes compared to the data set presented here. You have any idea why?

+ The new aspect relates to the “in situ” units used in this manuscript. It seems very difficult if not impossible to compare “in situ” volumes of gases in different depth or apply depth bins (e.g. 250 – 1000 m) to these numbers. The authors might want to choose units kg or mol, both of which are independent of water depth/pressure and commonly used in flux calculations and models. And change Fig. 7 accordingly.

+ e.g. Line 335 – 336 shelf contributes with 90% in terms of volume but 62% in terms of mass. Using volume has a distorting effect.

+ The importance of the shelf will decrease; which is important in the light of methane escaping into the atmosphere.

Detailed comments:

Abstract

Line 28: Expression: equivalent average methane flux-rate for Cascadia.. What does equivalent refer to? Cascadia is not defined in terms of depth range and longitudes/latitude and area.

Introduction

Line 47 - 49 "and oceanographic phenomena ... you might consider calling it oceanographic currents

Line 55 - 56 I do not quite understand the expression "migration pathways of fluids originating from the Tofino basin" compared to other fluid pathways observed?

Line 68 Instead of Cascadia you might want to consider calling it Cascadia margin as you do not refer to the mountain range.

Line 74 - 75: down to what depth?

Line 75: "fluid flow fluctuations" fluid includes gas and liquid - Do you mean disperse fluid flow compared to localized gas emissions??

Line 76: or the (w/o with)

Line 76 - 82: this is all very vague, very detailed and with little information: You might want to take it out or shorten it in the sense of:

... they provide indications of the flux variability over longer time scales suggesting possible influences (other than tidal) by earthquakes, seasonal upwelling or storm patterns.

Results

Fig 2 c and d: the heights of the flares could be indicated; Fig 3 b and c are mixed up somehow

The shelf environment:

Why did you take out the "70%"?

Line 154 You might want to take out "specific" as it is not defined.

To be a bit more positive I would put the positive things (15 years of observations) first before you start pointing out the shortcomings of the data set. This really is a great dataset!

Deep-water setting

The coverage is discuss but not quantified here - similar to the chapter on the shelf environment. It would be great if you could

Line 174 - 175 At this point the authors mention vents along the deformation front but miss out on some more details. I would suggest to delete this or be more specific as to why there are no flares to be expected?

Line 199 - 202: "... the shelf regions become more amplified (Fig. 4c)." Is this still true if you take out the high amount of vents in the JdF strait area?? If not it would rather be this particular environment that is amplified compared to the overall shelf... might be worth a try.

Line 198-99 "However, as the data footprint ..." delete "as"; Line 204: "by the small beam angles of the sounders used. ." (delete double dot at the end)

Figure 4: why do you use a different data set for Figures 4a and 4b? This does not seem to make sense in terms of comparability.

Acoustic quantification

Line 226 "flow estimates were carried out.." ; grammar: which were carried out or carried out..

Line 236 "in situ flow rates" ... inst oder intg??

Line 244- 245 "Here, we defined a bubble-size distribution using ROV video observations and estimated bubble rise speeds for a number of bubble sizes (see Methods)." The authors define one bubble-size distribution from one feeder channel and one depth range. All these variables change for any other vent site. Did the authors verify the model? They do later on – possibly refer to that chapter here. It might be worth a try to use a different model for the shelf sites only, e.g., using Veloso et al., 2015 or data listed within the publication (Fig. 11)

Line 241 -242 - "significant variations in the flow rate".. which are: numbers or %???

Line 243 "real changes in the flow" ... tidal changes are also real changes

Line 245-246 expression: "we added 310 flow-estimates at other flare locations"

Line 249 - 251: slope region near... define depth range

Line 252: in situ values (inst oder intg)

Line 252 ff: how does observation a) - the clustering at the JdF strait help with margin wide integration and defining average flow ratios? You might want to change the expression.

Line 260 ff: it would be more useful to convert into kg or mol before any comparison apart from the instantaneous flux possibly.

Margin-wide gas flux

Line 293-294 "and flow rates defined are representative for all systems (i.e. identical bubble-size distribution); do you mean: defined at one site and representative for all (see above)

Line 302 – 335 It is extremely difficult to follow the results chapter in this part with too many numbers being listed. You might want to consider rearranging (table/text).

Line 310 depth-dependent density of methane – no further clue as to how – cite Kossel et al.

Line 326 contributes ~11% to the margin-wide flow (measured in L/yr) - > this would make far more sense in mol or kg/yr since L at 200 m depth and 1000 m depth refer to very different amounts. Same is true for Lines 335 - 336!!! (90 → 62%) This is rather misleading.

Line 327 - 328: define the "the upper and lower bounds of the flow estimates"

Line 332-333 on the shelf?

Line 342 "...representing about 0.03% of the Earth's total seafloor area) could contribute between ~0.00067% and ~2.2% of the global seafloor methane emissions" What is it you want to point out with this?

Fig. 7 – I would suggest to use kg or mol for comparison

Discussion

Line 372 - canyons or head-scars (as they may cross sedimentary layering); morphology: ridges commonly have higher probability for venting;

Line 375 - occurred³⁹³⁹"; delete 39; Line 377 - "of crossing either the tide is not falling and emissions are reduced, or activity is at a non-active"; change to: venting is at a non-active...

Line 381-382 - where are these locations; point out in Fig. 7

Line 392 - 393 - add head scars and explain why - crossing layered sediments

Line 397 detecting

Line 410 - 414 "smear the acoustic return from smaller vent outlets into larger-looking flares. T... Thus, detecting flares using single beam data likely represents a lower limit in the number of emission sites." ; and an overestimation in terms of flux rates? Please discuss/mention

Line 415 define the normalization (foot print)

Line 432 - 433 "Applying our bubble-size distribution (Fig. 8) compared to other literature values from Svalbard⁵¹ yields instantaneous in situ flow rates that are smaller by a factor of 3"; Might Pakistan be a better choice (Römer et al., 2012) where data originate from a number of depths? Or an average of all? .. and please use mol or kg!!

Line 436 - 437 "Varying temperature within reasonable limits at flare sites changes the flow rate by up to 5%." What is the impact on conversion into mass?

Line 455 This might just be me being ignorant but I am surprised regarding the minimum total error of 5.4% if any estimation integrated into this number is higher than that.

Line 456 - 457 "456 However, the geographical pattern of flow rates may still be representative. Yet, some questions arise and require future attention:"; You might want to consider re-writing this part.

Line 465: delete "used"

Line 470 - 471 Please add range (error bar) since otherwise these numbers might well be used by others without your discussion on error.

Line 521 and Line 736 "We ignore the effect of seasonal variations and apply average values." On the Oregon shelf and upper slope these changes can be quite significant in terms of temperature (conversion of volume to mass) (just a thought)

Line 548 - 550 Which depths ranges are you referring to? 400 - 1200 m please mention.

Line 598 - 599 Those not familiar with these sites might appreciate a depth or rough idea of the whereabouts of these sites.

Line 640 - Volume or amount?

Line 715-716 – some of the parameters are also hard to define.

Line 717-718 “Here, we only include parameters that vary between flare sites and water depth... as well as assumptions made on bubble-size distribution” But you do not alter the bubble size distribution neither between flare sites nor water depth, do you?

Line 724 – You might want to use evaluate not “determine the uncertainty”.

Line 725 – 735 – I would like to suggest refining this discussion: Flux is higher when bubble size is larger; this does not seem very specific. Or ... it shows a general trend (and delete the rest).

Reply to comment made by reviewer #3

Overall, we appreciate the care and detail by which the reviewer is reading our manuscript, and many comments are useful to still improve the overall contribution. Below we outline the changes included into the manuscript and modifications applied to figures in response to the suggestions raised. However, we believe the major comments on more and different bubble size distributions are unwarranted. We already did include two different such distributions and went through the effort in evaluating the uncertainty in flow estimates that arises from these different distributions. We included data from a shallow water site (and cite it accordingly) and our own deep-water setting derived from the Clayoquot site off northern Cascadia. Searching for another Cascadia example (including the suggested data base at MBARI) has not been successful – mostly as the necessary calibration or size-references are unavailable in the (existing) video footages. We agree that it would be desirable if many more such distributions were available and included in our analyses. But at this point of time, they simply do not exist. Also, adding one or a couple more bubble-size distribution is not changing the outcome of our work and conclusions we can draw from the data we have available – the uncertainty will remain as large as it is while the reported mean values may indeed shift by a few percent – this is simply a result of the statistics of using 1100+ vents, their regional occurrences and the entire size of the margin over which we extrapolate the flow-rate estimates. One of the outcomes of our work is indeed that the bubble-size distributions may need more careful analyses and many more measurements in various settings could be an improvement – especially on the shelf. But that requires a significant amount of data acquisition and research and may or may not be successful in the implementation. Thus, we believe there is no reason for delaying the publication of our results any further.

In the previous versions of our manuscript, we already did use the suggestion by reviewer #3 on reporting the flow rates in not just volume (L/min or L/year) but also mass (kg/min or kg/year). After these additional comments, we went through the manuscript again and modified text and figures to show both volume- and mass-based comparisons throughout (e.g. updated Figures 6 and 7).

In reply to the issue of geographical distribution of vent sites, we would like to highlight that the full statistical treatment of the data is well beyond the scope of this initial contribution. Reviewer #3 may be thinking of a boots-trap technique when suggesting leaving out a certain geographical region and re-defining the methane flux. While this is probably one good approach to go about this issue, we believe that it would make the story even more complicated, too long and convoluted, and at this point no further material, analyses and techniques need to be included into the manuscript.

Below we report individual responses to the individual comments made.

+ Random versus geographically controlled distribution: the “Results” mirrors this issue. While one paragraph discusses the clustering of venting the next paragraph basically states that this is interesting to know but that a random pattern is applied. Even if it does not erase the problem you might want to separate the two topics. It makes it hard to believe in the calculations and it will not help discussing on the causes of the vent distribution.

➔ *See general comments made above*

+ Line 199 - 202: "... the shelf regions become more amplified (Fig. 4c)." Is this also true if you ignore the large amount of vents in the JdF strait area?? Is it this particular environment that is amplified or is it the overall shelf. Might be worth a try? How does this relate to the 500 m isobar?

➔ *See general comments made above. There is no link to the 500-m isobar, which is not a significant depth cut-off limit in the context of this discussion of using (or excluding) a specific geographic region.*

+ Line 394 - 399 I cannot quite make out the argument here. As it seems to me, the authors mention a method for random distribution, do not use it and conclude that there is a random distribution. As it is this argument /paragraph does not seem convincing.

Yes, the flare distribution is assumed to be random, but so was the data acquisition, and so we cannot apply the known method which, as we wrote, fails the underlying criteria in the method. This paragraph was added to briefly address this problem and guide future missions to be more systematic with regular sampling grids and times, so we would like to keep it.

+ Line 415 – 423 "reveals that 2/3 of these flares are occurring between 47.5° and 48.25° latitude." The authors have already commented on this.

➔ *No changes required*

+ The bubble size issue: Is there no data set (MBARI?) for any shallower vent site. E.g. Lines 335 – 336, Line 432 - 433. Regarding the large difference in pressure and change in relative pressure between the deep sites and the shallow sites, it is very unlikely that bubble size maxima are similar in both environments. Referring to Fig. 11 in Veloso et al., 2015, the bubble size maxima at Clayoquot of 1 – 3 mm are low compared to any of the other sites shown in this figure. The Santa Barbara Channel site might be rather shallow and maxima are slightly higher; the Svalbard data show 4 -6 mm maxima, however, there are thermogenic sources as well. Why choose this data set and use it for all depth not for the shelf only? Anything above 4 mm significantly changes the bubble rise speed (your Fig. 8c).

➔ *See general comments made above. We included two drastically different bubble-size distributions and carried out analyses for both independently to evaluate the range of flow rates. Our combined discussion on uncertainty already honors this problem reviewer #3 is referring to. Choosing yet another different shallow water distribution would not add any new value to the discussion. Only by having dozens of such distributions for different water depths and different geographical regions would allow a better treatment of the flow rate estimation and regional extrapolation. Yet, such data are years if not decades away from being available (if ever).*

+Would it be worthwhile to calculate the shelf data using one of the shallower data sets while keeping your data for the deep vents?

➔ *See general comments made above – this boots-trap technique is one option on evaluating the statistics of the data.–We restrain from any more detailed work at this point as it would be outside the scope of the current contribution and also would unnecessarily delay the publication of our work.*

+In Veloso's paper there are some other bubble size distributions all of which have higher average bubble sizes compared to the data set presented here. You have any idea why?

➔ *This is indeed interesting but no, we do not know why the bubble sizes are different. Discussions to the cause of the differences are outside the scope of our contribution anyway.*

+ The new aspect relates to the “in situ” units used in this manuscript. It seems very difficult if not impossible to compare “in situ” volumes of gases in different depth or apply depth bins (e.g. 250 – 1000 m) to these numbers. The authors might want to choose units kg or mol, both of which are independent of water depth/pressure and commonly used in flux calculations and models. And change Fig. 7 accordingly.

➔ *See general comments made above. The original manuscript already had included mass-based flow rates. However, we did change though Figures 6 and 7 to include the statistics on mass-based flow rates following the reviewer's comments.*

+ e.g. Line 335 – 336 shelf contributes with 90% in terms of volume but 62% in terms of mass. Using volume has a distorting effect.

➔ *See general comments made above*

+ The importance of the shelf will decrease; which is important in the light of methane escaping into the atmosphere.

➔ *Based on earlier comments made by all the reviewer's we restrain from discussing the fate of methane, escape into the atmosphere, and any linkages to climate change.*

Detailed comments:

Abstract

Line 28: Expression: equivalent average methane flux-rate for Cascadia.. What does equivalent refer to? Cascadia is not defined in terms of depth range and longitudes/latitude and area.

➔ *Text-style was modified to address this comment.*

Introduction

Line 47 - 49 “and oceanographic phenomena ... you might consider calling it oceanographic currents

➔ *Text was modified to address this comment. Currents are just one of these phenomena, including upwelling.*

Line 55 - 56 I do not quite understand the expression "migration pathways of fluids originating from the Tofino basin" compared to other fluid pathways observed?

➔ *Text-style was modified to address this comment. Migration of fluids along conduits from the Tofino basin to Barkley canyon were suggested previously using Strontium isotopes, but those pathways were not seismically imaged.*

Line 68 Instead of Cascadia you might want to consider calling it Cascadia margin as you do not refer to the mountain range.

➔ *Agreed, the text was modified to address this comment.*

Line 74 - 75: down to what depth?

➔ *We added text to address this comment.*

Line 75: "fluid flow fluctuations" fluid includes gas and liquid - Do you mean disperse fluid flow compared to localized gas emissions??

➔ *Text was modified to address this comment. In the context of OSMO-samplers, we mean pore-fluids (no gas).*

Line 76: or the (w/o with)

➔ *Text was modified to address this comment.*

Line 76 - 82: this is all very vague, very detailed and with little information: You might want to take it out or shorten it in the sense of: ... they provide indications of the flux variability over longer time scales suggesting possible influences (other than tidal) by earthquakes, seasonal upwelling or storm patterns.

➔ *We actually did not modify this section too much, as we believe that the length of text is appropriate and adequate. This is the only section where we mention other, non-acoustic, long-term data related to ocean observatories that help understand the overall dynamics and variability of the shallow fluid system, and we anticipate readers to be curious what else is out there that could have helped this study, so we also need to explain that these other data have not yet produced results useful for our study.*

Results

Fig 2 c and d: the heights of the flares could be indicated; Fig 3 b and c are mixed up somehow

- ➔ *Each sub-figure does have an indicator for height. As we do not statistically treat in any way the actual gas flare extent (height in water column) we see no need to add this level of detail to one specific figure as it will never be followed up upon in our analyses. Fig. 3a and 3b are corrected in the caption.*

The shelf environment:

Why did you take out the "70%"?

- ➔ *Good question, no real reason why not to keep it. We probably did this due to a final editing effort to reduce number of words to the cut-off value of the journal. We added it back in.*

Line 154 You might want to take out "specific" as it is not defined.

- ➔ *Agreed, text was modified.*

To be a bit more positive I would put the positive things (15 years of observations) first before you start pointing out the shortcomings of the data set. This really is a great dataset!

- ➔ *Text-style was modified to address this comment.*

Deep-water setting

The coverage is discussed but not quantified here - similar to the chapter on the shelf environment.

It would be great if you could Line 174 - 175 At this point the authors mention vents along the deformation front but miss out on some more details. I would suggest to delete this or be more specific as to why there are no flares to be expected?

- ➔ *Text was modified to address this comment. There are few crossings from which we could attempt identifying vents (and found very few if any) and historically there are very few observations to start with. We kept the discussion but modified the sentences a bit.*

Line 199 - 202: "... the shelf regions become more amplified (Fig. 4c)." Is this still true if you take out the high amount of vents in the JdF strait area?? If not it would rather be this particular environment that is amplified compared to the overall shelf... might be worth a try.

- ➔ *See general comments above on such boots-trapping; we do not change the text nor make any additional more complicated analyses.*

Line 198-99 "However, as the data footprint ..." delete "as";

- ➔ *Text was modified to address this comment.*

Line 204: "by the small beam angles of the sounders used. ." (delete double dot at the end)

- ➔ *Text was modified to address this comment.*

Figure 4: why do you use a different data set for Figures 4a and 4b? This does not seem to make sense in terms of comparability.

➔ *The reason is that we have two types of observations of gas flares: (a) from our own picking in acoustic data with navigation available and (b) historical information, mostly without the accompanying acoustic data and navigation (but we know if it was multi- or single-beam). Thus, showing the depth-distribution along ship tracks (Figure 4a) can only be made for data where acoustic data are available (not including the historic information); Figure 4b then includes all vents from all sources, but lacking info on the exact track-line navigation from which the historical data were derived, we use the general bathymetry (depth). Figure 4c then can exploit the knowledge of sounder-type to estimate the footprint (single/multibeam) to normalize the data for all vents. However, after additional thoughts and literature search to find the acoustic sources of historical vent observations, we removed sub-panel (a) and only keep a comparison between all acoustically observed vents normalized by depth of bathymetry (as done by other authors) and acoustic footprint where we were able to calculate it.*

Acoustic quantification

Line 226 "flow estimates were carried out.." ; grammar: which were carried out or carried out..

➔ *Text-style was modified to address this comment.*

Line 236 "in situ flow rates" ... inst oder intg??

➔ *Text was modified to address this comment. Basically all types of flow rates.*

Line 244- 245 "Here, we defined a bubble-size distribution using ROV video observations and estimated bubble rise speeds for a number of bubble sizes (see Methods)." The authors define one bubble-size distribution from one feeder channel and one depth range. All these variables change for any other vent site. Did the authors verify the model? They do later on – possibly refer to that chapter here. It might be worth a try to use a different model for the shelf sites only, e.g., using Veloso et al., 2015 or data listed within the publication (Fig. 11)

➔ *See general comments above about including other distributions and the impact that may have on the uncertainty evaluation.*

Line 241 -242 - "significant variations in the flow rate".. which are: numbers or %???

➔ *In order to make life simple, we remove the hint to uncertainty here (as it is discussed in great detail later in the manuscript) and only draw on the generalized observations.*

Line 243 "real changes in the flow" ... tidal changes are also real changes

➔ *Text-style was modified to address this comment.*

Line 245-246 expression: "we added 310 flow-estimates at other flare locations"

➔ *Text-style was modified to address this comment.*

Line 249 - 251: slope region near... define depth range

➔ *Text-style was modified to address this comment.*

Line 252: in situ values (inst oder intg)

➔ *Text-style was modified to address this comment.*

Line 252 ff: how does observation a) - the clustering at the JdF strait help with margin wide integration and defining average flow ratios? You might want to change the expression.

➔ *See general comments above on regional data coverages.*

Line 260 ff: it would be more useful to convert into kg or mol before any comparison apart from the instantaneous flux possibly.

➔ *See general comments above. We added where not already included previously the mass-based flow rate estimates.*

Margin-wide gas flux

Line 293-294 “and flow rates defined are representative for all systems (i.e. identical bubble-size distribution); do you mean: defined at one site and representative for all (see above)

➔ *Yes; no need to change text*

Line 302 – 335 It is extremely difficult to follow the results chapter in this part with too many numbers being listed. You might want to consider rearranging (table/text).

➔ *We reduced the text, added a new table (now Table 4) and hope that the results are now easier to digest.*

Line 310 depth-dependent density of methane – no further clue as to how – cite Kossel et al.

➔ *The citation was already included in Table 3 and mentioned in the Methods – we moved it also into the text to address this comment.*

Line 326 contributes ~11% to the margin-wide flow (measured in L/yr) - > this would make far more sense in mol or kg/yr since L at 200 m depth and 1000 m depth refer to very different amounts. Same is true for Lines 335 - 336!!! (90 → 62%) This is rather misleading.

➔ *See general comments above. We added those mass-based values where not already done before.*

Line 327 - 328: define the "the upper and lower bounds of the flow estimates"

➔ *Text was modified to address this comment.*

Line 332-333 on the shelf?

➔ *No, the total average only. We then go on and differentiate in shelf/deep water etc. No need to change text.*

Line 342 “...representing about 0.03% of the Earth’s total seafloor area) could contribute between ~0.00067% and ~2.2% of the global seafloor methane emissions" What is it you want to point out with this?

➔ *Basically, that our estimates are reasonable and within the expected range given uncertainty and size of the margin.*

Fig. 7 – I would suggest to use kg or mol for comparison

➔ *See discussion and general comments made above. We show now the integrated mass values.*

Discussion

Line 372 - canyons or head-scars (as they may cross sedimentary layering); morphology: ridges commonly have higher probability for venting;

➔ *Text-style was modified to address this comment.*

Line 375 - occurred3939"; delete 39;

➔ *Text-style was modified to address this comment.*

Line 377 - "of crossing either the tide is not falling and emissions are reduced, or activity is at a non-active"; change to: venting is at a non-active...

➔ *Text-style was modified to address this comment. But there is a difference from vent-inactivity due to the time relative to the tide and inactivity because of long-term fluctuations.*

Line 381-382 - where are these locations; point out in Fig. 7

➔ *Labels are now added to figure.*

Line 392 - 393 - add head scars and explain why - crossing layered sediments

➔ *Text-style was modified to address this comment.*

Line 397 detecting

➔ *Text-style was modified to address this comment.*

Line 410 - 414 "smear the acoustic return from smaller vent outlets into larger-looking flares. T... Thus, detecting flares using single beam data likely represents a lower limit in the number of emission sites." ; and an overestimation in terms of flux rates? Please discuss/mention

➔ *No, not really. If the single beam data combine ('smear') contributions of 5 vents into one value, the combined flow rate is higher. Yet, each individual flare, if taken isolated, is showing a lower flow-rate value. Yet, the summation remains the same (i.e. $1 * 500 = 5 * 100$). This is one of the problems that extrapolation suffers from; any boots-trapping or other clever statistical treatment (Horvitz-Thompson estimators) must be able to deal with this issue – a yet unresolved problem.*

Line 415 define the normalization (foot print)

➔ *Text-style was modified to address this comment.*

Line 432 - 433 "Applying our bubble-size distribution (Fig. 8) compared to other literature values from Svalbard51 yields instantaneous in situ flow rates that are smaller by a factor of 3"; Might Pakistan be a better choice (Römer et al., 2012) where data originate from a number of depths? Or an average of all? .. and please use mol or kg!!

➔ *See comments made above on adding another distribution and (non-existing) impact on the results/outcome.*

Line 436 - 437 "Varying temperature within reasonable limits at flare sites changes the flow rate by up to 5%." What is the impact on conversion into mass?

➔ *This question is at a very detailed level of the physics. Density is changing with water depth as implemented in our analyses. If volume goes up by 5%, then the conversion to mass is also changed by 5%. Only the absolute number measured in kg is different between shallow and deep water.*

Line 455 This might just be me being ignorant but I am surprised regarding the minimum total error of 5.4% if any estimation integrated into this number is higher than that.

➔ *We actually mean that the lower bound is 5.4% (actually 4.5% in the newly revised version) of the reported mean and the upper bound is 820% (actually 1800% in the newly revised version). E.g., if the mean flow rate were to be 100 L/min, the lower bound is 5.4 L/min, and the upper bound is 820 L/min.*

Line 456 - 457 "456 However, the geographical pattern of flow rates may still be representative. Yet, some questions arise and require future attention:"; You might want to consider re-writing this part.

➔ *We are not sure why we should rewrite this portion and discussion.*

Line 465: delete "used"

➔ *Text was modified to address this comment.*

Line 470 - 471 Please add range (error bar) since otherwise these numbers might well be used by others without your discussion on error.

➔ *Text was modified to address this comment.*

Line 521 and Line 736 "We ignore the effect of seasonal variations and apply average values." On the Oregon shelf and upper slope these changes can be quite significant in terms of temperature (conversion of volume to mass) (just a thought)

➔ *No need for any changes.*

Line 548 - 550 Which depths ranges are you referring to? 400 - 1200 m please mention.

➔ *Text was modified to address this comment. Taking from the Figure 8b.*

Line 598 - 599 Those not familiar with these sites might appreciate a depth or rough idea of the whereabouts of these sites.

➔ *Text was modified to address this comment.*

Line 640 - Volume or amount?

➔ *Text-style was modified to address this comment: "Volume"*

Line 715-716 – some of the parameters are also hard to define.

➔ *No need for any changes.*

Line 717-718 “Here, we only include parameters that vary between flare sites and water depth... as well as assumptions made on bubble-size distribution” But you do not alter the bubble size distribution neither between flare sites nor water depth, do you?

➔ *See comments above: we do make two full calculations for two different size distributions and carry out an uncertainty evaluation from those values.*

Line 724 – You might want to use evaluate not “determine the uncertainty”.

➔ *Text was changed accordingly.*

Line 725 – 735 – I would like to suggest refining this discussion: Flux is higher when bubble size is larger; this does not seem very specific. Or ... it shows a general trend (and delete the rest).

➔ *See comments above: we compare two distributions and compare the outcome of flow rate. Thus, only generalized (simple) observations are made – we like to keep the discussion as it is and not make things unnecessarily complex.*